# BubR1 alterations that reinforce mitotic surveillance act against aneuploidy and cancer

Robbyn L Weaver[1], Jazeel F Limzerwala[1], Ryan M Naylor[1], Karthik B Jeganathan[2], Darren J Baker[1,2], Jan M van Deursen[1,2]*

[1]Department of Biochemistry and Molecular Biology, Mayo Clinic, Rochester, United States; [2]Department of Pediatric and Adolescent Medicine, Mayo Clinic, Rochester, United States

**Abstract** BubR1 is a key component of the spindle assembly checkpoint (SAC). Mutations that reduce BubR1 abundance cause aneuploidization and tumorigenesis in humans and mice, whereas BubR1 overexpression protects against these. However, how supranormal BubR1 expression exerts these beneficial physiological impacts is poorly understood. Here, we used *Bub1b* mutant transgenic mice to explore the role of the amino-terminal (BubR1$^N$) and internal (BubR1$^I$) Cdc20-binding domains of BubR1 in preventing aneuploidy and safeguarding against cancer. BubR1$^N$ was necessary, but not sufficient to protect against aneuploidy and cancer. In contrast, BubR1 lacking the internal Cdc20-binding domain provided protection against both, which coincided with improved microtubule-kinetochore attachment error correction and SAC activity. Maximal SAC reinforcement occurred when both the Phe- and D-box of BubR1$^I$ were disrupted. Thus, while under- or overexpression of most mitotic regulators impairs chromosome segregation fidelity, certain manipulations of BubR1 can positively impact this process and therefore be therapeutically exploited.

**\*For correspondence:**
vandeursen.jan@mayo.edu

**Competing interests:** The authors declare that no competing interests exist.

## Introduction

Chromosomal instability (CIN) describes a condition where cells frequently acquire cytogenetic alterations and do not accurately segregate their chromosomes (*Giam and Rancati, 2015*). Aneuploidy, defined as a state in which there are alterations to whole chromosome copy number, results from CIN and is a feature of almost all tumors, but whether aneuploidy is a cause or consequence of transformation is the subject of much debate (*Ricke and van Deursen, 2013*). CIN is thought to allow pre-neoplastic cells to acquire genes that promote tumor progression and lose those which suppress transformation (*Baker et al., 2009*; *Burrell et al., 2013*; *Hanahan and Weinberg, 2011*; *Loeb, 2011*) and there are multiple lines of evidence which support aneuploidy having a causative role for cancer. For instance, several human aneuploidy syndromes are characterized by increased susceptibility to cancer, including trisomies 8, 18 (Edwards syndrome) and 21 (Down Syndrome) (*Ganmore et al., 2009*), and mosaic variegated aneuploidy (MVA) (*Hanks et al., 2004*; *Snape et al., 2011*). Furthermore, bidirectional deviations in protein levels of various mitotic regulators, including Mad2, Mad1 and Bub1, cause aneuploidy and tumor predisposition in mice (*Iwanaga et al., 2007*; *Jeganathan et al., 2007*; *Michel et al., 2001*; *Ricke et al., 2011*; *Ryan et al., 2012*; *Sotillo et al., 2007*). Additionally, modulations to a spectrum of other proteins that participate in diverse cellular functions, such as the E2 ubiquitin-conjugating enzyme Ubch10 (*van Ree et al., 2010*), the centromere-linked microtubule protein CENP-E (*Weaver et al., 2007*), and the nuclear pore complex protein Nup88 (*Naylor et al., 2016*) result in aneuploidy and accelerated cancer progression. Finally,

**eLife digest** Human DNA is organized into 46 chromosomes, which must be duplicated before a cell divides and are then shared equally between the two new cells. When this process goes awry, the new cells either have too many or too few chromosomes. This situation – known as aneuploidy – frequently occurs in cancer cells, and is thought to cause cells to gain extra copies or lose copies of genes that promote or prevent cancer, respectively.

Cells have several ways to prevent aneuploidy. One of these safeguards, known as the spindle assembly checkpoint (SAC), involves a protein called BubR1, which acts at the stage when the duplicated chromosomes need to be equally divided into each daughter cell. Mouse models show that low levels of the BubR1 protein result in aneuploidy and increased predisposition to cancer. High levels of BubR1, on the other hand, allow the mice to stay healthier for longer and can stop tumors from forming. However, it was not known exactly how high amounts of BubR1 protect against cancer.

To address this question, Weaver et al. set out to determine which parts, or domains, of the BubR1 protein protect against cancer. Mice with high levels of the full-length BubR1 protein were compared with mice that made mutant versions of BubR1 lacking certain domains. These experiments revealed that a small portion of the beginning of the protein was necessary to protect against tumor formation, but removing a large region in the middle of BubR1 still protected mice against lung cancer and aneuploidy. Additional experiments performed on mouse cells grown in the laboratory revealed that whole BubR1 protein and the mutant protein lacking the middle region might prevent aneuploidy in multiple ways. For example, both systems had stronger SAC signaling, which could serve to make segregating the chromosomes more accurate.

In the future, it will be important to find out whether BubR1 acts in the same way in human cells and cancers. Lastly, since it is not possible to over-produce BubR1 in humans, other methods will need to be investigated to use this knowledge to treat cancer.

genome-wide screens of proteins that negatively (STOP) and positively (GO) regulate proliferation are recurrently and selectively lost and gained respectively in either focal regions or whole chromosomes (*Davoli et al., 2013*; *Solimini et al., 2012*). This suggests a model where changes in gene copy number are under selection rather than simply accompanying transformation, supporting genomic instability as a driver of cancer (*Davoli et al., 2013*; *Solimini et al., 2012*). On the other hand, several other mouse models of CIN have revealed inconsistent results regarding the relationship between aneuploidy and cancer, where some mouse models with elevated levels of aneuploidy do not show increased susceptibility to cancer (*Babu et al., 2003*; *Kalitsis et al., 2005*; *Ricke et al., 2012*). Furthermore, proteotoxic stress from increased gene expression in cells with extra chromosome copies has adverse effects on cell growth and may thus counteract cancer progression (*Tang et al., 2011*; *Williams et al., 2008*).

Aneuploidy results when cells fail to segregate chromosomes properly. To promote high-fidelity separation of duplicated chromosomes, cells have the machinery to safeguard against missegregation. One such mechanism is the spindle assembly checkpoint (SAC). This surveillance system prevents activation of the E3 ubiquitin ligase anaphase-promoting complex/cyclosome (APC/C) by its co-activator, Cdc20 (*Peters, 2006*). This ensures chromosomal stability by preventing sister chromatid separation prior to bi-orientation of mitotic chromosomes at the metaphase plate (*Musacchio and Salmon, 2007*). An additional measure to promote accurate chromosome segregation is allowing sufficient time to form proper and correct erroneous kinetochore-microtubule (MT-KT) attachments prior to anaphase onset (*Meraldi et al., 2004*). Merotely, a type of improper attachment in which a single kinetochore is attached to microtubules emanating from both spindle poles, is undetected by the SAC and can result in lagging chromosomes (*Cimini et al., 2001*; *Rodriguez-Bravo et al., 2014*). Mad1/2, Mps1, and BubR1 specify the minimum time in mitosis, and loss of these proteins reduces the duration of mitosis and increases the rates of missegregation (*Maciejowski et al., 2010*; *Meraldi et al., 2004*; *Rodriguez-Bravo et al., 2014*).

BubR1, along with Mad2 and Bub3, is a component of the mitotic checkpoint complex (MCC), which mediates the SAC (*Kulukian et al., 2009*; *Musacchio and Salmon, 2007*; *Sudakin et al., 2001*). Once each chromosome has properly and stably attached to the mitotic spindle and sufficient inter-kinetochore tension is generated, the MCC dissociates, allowing Cdc20 to activate the APC/C (*Musacchio and Salmon, 2007*). Co-activation of APC/C by Cdc20 in metaphase results in the polyu-biquitination and subsequent proteasomal degradation of cyclin B1 and securin, thereby triggering sister chromatid separation and anaphase onset (*Kapanidou et al., 2015*; *Musacchio and Salmon, 2007*). BubR1, encoded by the gene *Bub1b* in mice or *BUB1B* in humans, is a modular protein, with several known functional domains that together ensure mitotic fidelity and genome stability. BubR1 localizes to the kinetochore by interacting through its GLEBs-like motif with Bub3 (*Elowe et al., 2010*; *Lampson and Kapoor, 2005*). In human cells, kinetochore-localized BubR1 was shown to be important for MT-KT stabilization through an internally located kinetochore attachment and regula-tory domain (KARD) (*Suijkerbuijk et al., 2012*). The KARD allows kinetochore localization of the phosphatase PP2A, which counteracts the MT-KT destabilizing activity of Aurora B kinase, a key mediator of error-correction (*Ruchaud et al., 2007*; *Suijkerbuijk et al., 2012*). Additional BubR1 functional domains include a putative kinase/pseudokinase domain that has been reported to rein-force the SAC and stabilize MT-KT attachments (*Elowe, 2011*; *Harris et al., 2005*; *Suijkerbuijk et al., 2012*) and two Cdc20-binding domains, of which the N-terminal domain (BubR1$^N$) is a critical APC/C$^{Cdc20}$ inhibitor essential for cell survival (*Malureanu et al., 2009*). BubR1$^N$ contains two KEN-boxes, spanning amino acids 19–21 (KEN1) and 298–300 (KEN2) in mice, that in conjunction with a destruction (D)-box (D1) just downstream of KEN2 permits BubR1 to behave as a pseudo-substrate inhibitor of APC/C$^{Cdc20}$(*Burton and Solomon, 2007*; *Chao et al., 2012*; *Han et al., 2013*; *Izawa and Pines, 2015*; *Malureanu et al., 2009*). Recent work has highlighted that BubR1 is capable of binding both soluble Cdc20 through KEN1 to prevent APC/C-Cdc20 associ-ation, and a second Cdc20 that has already bound to and activated the APC/C through a combina-tion of KEN2 and D1 for even more dynamic APC/C$^{Cdc20}$ inhibition (*Izawa and Pines, 2015*).

The second Cdc20-binding domain of BubR1, BubR1$^I$, is an internally located and functionally dis-tinct region important for Cdc20 kinetochore recruitment, and has been proposed to serve a dual function in SAC activation and silencing (*Chao et al., 2012*; *Di Fiore et al., 2015*; *Diaz-Martinez et al., 2015*; *Izawa and Pines, 2015*; *Lischetti et al., 2014*; *Tang et al., 2001*). Several conserved and somewhat redundant motifs within BubR1$^I$ were recently identified that are thought to promote the BubR1-Cdc20 interaction through complementary mechanisms: the ABBA motif, named for its conserved presence in <u>A</u>cm1, <u>B</u>ub1, <u>B</u>ubR1 and Cyclin <u>A</u> (*Di Fiore et al., 2015*); the Phe box, a phenylalanine-containing region which is encompassed within the ABBA motif (*Diaz-Martinez et al., 2015*); and a D-box just downstream of the Phe Box (D-box2) (*Diaz-Martinez et al., 2015*).

Whereas bidirectional changes to protein levels of Mad1, Bub1 and Mad2 cause aneuploidy and tumorigenesis (*Ricke and van Deursen, 2013*), BubR1 is unique amongst mitotic regulators in that both under- and overexpression results in drastically different phenotypes (*Baker et al., 2013*; *Baker et al., 2004*). Complete loss of BubR1 causes early embryonic death (*Wang et al., 2004*), and while BubR1 hypomorphic (*Bub1b$^{H/H}$*) mice are viable, they develop a variety of premature aging phenotypes (*Baker et al., 2004*; *Hartman et al., 2007*; *Kyuragi et al., 2015*; *Matsumoto et al., 2007*; *North et al., 2014*), progressive aneuploidy (*Baker et al., 2004*), and are predisposed to car-cinogen-induced cancers (*Baker et al., 2006*). Additionally, in humans, mutations in *BUB1B* have been causally implicated in MVA, a rare clinical syndrome characterized by widespread aneuploidy, growth retardation, shortened lifespan, and cancer predisposition (*García-Castillo et al., 2008*; *Hanks et al., 2004*; *Matsuura et al., 2006*; *Wijshake et al., 2012*). Conversely, overexpression of BubR1 extends life- and healthspan of mice, decreases the tumor incidence, and provides protection against age-related phenotypes in tissues that are prone to increased aneuploidy rates with age (*Baker et al., 2013*).

Despite profound anti-tumor and anti-aneuploidization effects of BubR1 overexpression, the molecular mechanism(s) of how it prevents CIN and cancer remains unclear (*Baker et al., 2013*). Here, we focus on the role of BubR1-Cdc20 binding, and explore how this interaction reinforces the SAC and error-correction machinery by using a series of transgenic mice overexpressing BubR1 mutants with disruptions in Cdc20-binding domains. We show that overexpression of a mutant BubR1 that includes disruptions of the internal Cdc20-binding domain (BubR1$^{\Delta I}$) elicits a tumor-

protective mechanism similar to that of full-length (FL)-BubR1 overexpression. Importantly, like in FL-*Bub1b*, overexpression of this mutant also safeguards against aneuploidization, likely by both strengthening SAC signaling and preventing improper KT-MT attachments. Thus, the internal Cdc20-binding domain is dispensable to mediate these protective effects, while the N-terminal Cdc20-binding domain is necessary, but not sufficient. BubR1$^{ΔI}$ also provides distinct molecular properties unique to that of overexpression alone that also likely promote genetic stability and show no overt detrimental effects on cells or mice. This includes a more robust SAC that is more responsive to weak stimuli, and an increase in the normal length of mitosis. With further refined mutant *Bub1b* constructs, we demonstrate that a maximal SAC response can be achieved exclusively by the loss of the Phe box and D-box2, and that the mitotic timing may be dependent on previously uncharacterized regions of BubR1. Importantly, this work sheds light on the causal role of CIN in cancer by demonstrating that enhancing genomic stability fortifies the barriers of transformation, and may provide unique insights into the generation of new therapeutic strategies.

## Results

### Generation of transgenic mice overexpressing BubR1 mutants

To determine the role of the N-terminal and internal Cdc20-binding domains in the protective effect of BubR1 overexpression on aneuploidy and cancer, we generated three distinct Flag-tagged mutant *Bub1b* transgenic mouse strains (*Figure 1A*). The first two mutants lacked either residues 1–363 containing the N-terminal Cdc20-binding domain (*Bub1b*$^{ΔN}$) or residues 525–700 (*Bub1b*$^{ΔI}$) which disrupts the Phe box, and removes D-box2 and KARD. The third mutant contained only the N-terminal Cdc20-binding domain (*Bub1b*$^{N}$). Like FL-*Bub1b*, all three mutants were expressed under the control of the ubiquitously active CAAGS promoter (*Baker et al., 2013*). Enhanced green fluorescent protein (EGFP) was co-expressed from an internal ribosome entry site (IRES). Western blots of mouse embryonic fibroblasts (MEFs) and lung tissue from 5-month old mice revealed that each of the three BubR1 mutants was expressed at levels comparable to that of FL-BubR1 (strain T23; *Figure 1B*, and *Figure 1—figure supplement 1*).

BubR1 is unique among mitotic regulators in that its overexpression does not lead to chromosome missegregation and aneuploidization and actually protects cells against chromosomal instability and karyotypic abnormalities (*Baker et al., 2013*; *Ricke et al., 2011*; *Ryan et al., 2012*; *Sotillo et al., 2007*). To examine whether the *Bub1b* mutants we created negatively impacted karyotype integrity, we performed chromosome counts on metaphase spreads of MEFs derived from wild-type and *Bub1b* transgenic MEFs (*Table 1*). There was no significant difference in aneuploidy rates between FL-*Bub1b*, *Bub1b*$^{N}$, *Bub1b*$^{ΔI}$ and wild-type MEFs, whereas *Bub1b*$^{ΔN}$ MEFs had increased aneuploidy. However, these aneuploidy-prone MEFs did not have higher rates of chromosome segregation errors as assessed by live cell imaging (*Table 2*). As expected, missegregation rates for the FL-*Bub1b*, *Bub1b*$^{N}$, and *Bub1b*$^{ΔI}$ mutants were normal. By interphase FISH, none of the transgenic mouse lines, including *Bub1b*$^{ΔN}$, showed evidence of elevated aneuploidy rates in a broad spectrum of mouse tissues and organs collected from 5-month-old mice (*Table 3*). Altogether, these data indicated that our transgenic mutant lines could provide the framework necessary to characterize the benefits of FL-BubR1 overexpression.

### BubR1 N-terminus is necessary but not sufficient to protect against cancer and aneuploidy

In earlier studies, we found that overexpression of FL-BubR1 markedly inhibits lung tumor formation in *Kras*$^{LA1}$ mice, a genetically engineered strain carrying a conditional oncogenic *Kras* allele (*Kras*$^{G12D}$) that becomes active upon intrachromosomal homologous recombination (*Baker et al., 2013*; *Johnson et al., 2001*). Given the robustness of this tumor protection, we used *Kras*$^{LA1}$ mice to explore the role of the amino-terminal and internal Cdc20-binding domains in the tumor protective effect of BubR1 overexpression on cancer. Consistent with our previously published data (*Baker et al., 2013*), overexpression of FL-BubR1 had a tumor-protective effect (*Figure 2A–C*). *Bub1b*$^{N}$ and *Bub1b*$^{ΔN}$, however, were unable to ameliorate the tumor burden of *Kras*$^{LA1}$ mice, indicating that binding of Cdc20 mediated by the N-terminal domain is necessary, but not sufficient, to protect against tumor formation. In contrast, *Bub1b*$^{ΔI}$ fully retained the tumor-protective benefit of

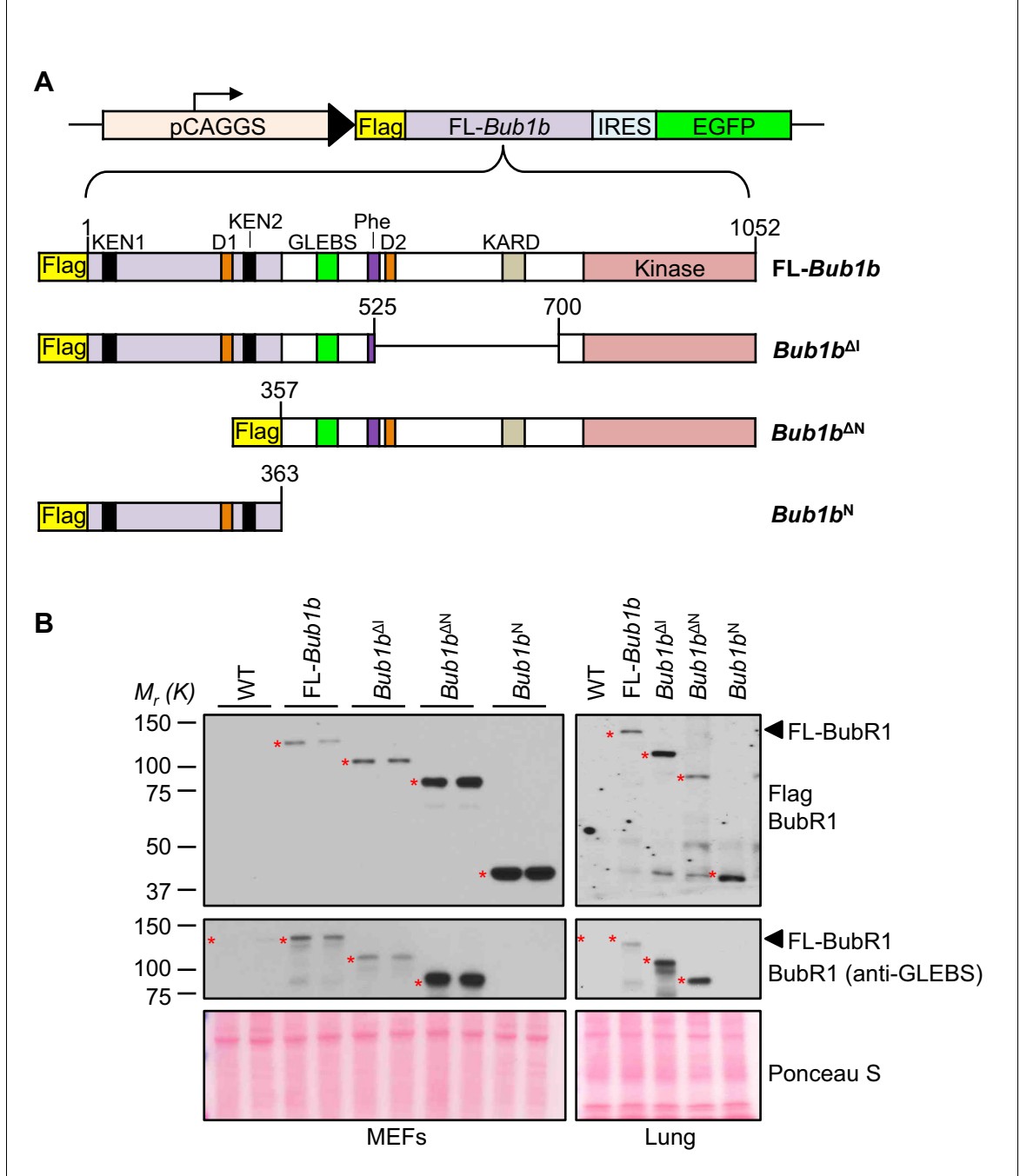

**Figure 1.** BubR1 transgenic mutant proteins are highly overexpressed in vitro and in vivo. (**A**) (top) *Bub1b* transgenic vector design. pCAGGS, promoter consisting of the CMV immediate enhancer and the chicken-actin promoter. FL, full-length. IRES, internal ribosome entry site. (bottom) Schematics of the Flag-*Bub1b* transgenic mouse constructs. KEN, KEN-box. D, destruction-(D-)box. GLEBs, GLEBs-binding motif. Phe, Phe box. KARD, kinetochore attachment regulatory domain. (**B**) Western blots of MEF (left) and lung tissue extracts (right) from wild-type (WT) and Flag-*Bub1b* transgenic mice. Blots were probed with the indicated antibodies. Ponceau S was used to normalize loading.

The following figure supplement is available for figure 1:

**Figure supplement 1.** Analysis of BubR1 overexpression in transgenic MEFs.

**Table 1.** Karyotypes are stable in select BubR1 mutant overexpressing MEFs. Karyotype analysis of passage 5 MEFs of indicated genotype. $n \geq 3$ lines, 50 cells per line. Data are mean ± s.d. WT, wild-type. FL, full-length. (See associated **Table 1— source data 1**).

| Mitotic MEF genotype (n) | Mitotic figures | Aneuploid figures % (s.d) | Karyotype with indicated chromosome number | | | | | | | | |
| --- | --- | --- | --- | --- | --- | --- | --- | --- | --- | --- | --- |
| | | | 36 | 37 | 38 | 39 | 40 | 41 | 42 | 43 | 44 |
| WT (5) | 250 | 9 (6) | 1 | 1 | 0 | 5 | 227 | 9 | 7 | 0 | 0 |
| FL-*Bub1b* (3) | 150 | 10 (3) | 0 | 0 | 1 | 2 | 135 | 7 | 5 | 0 | 0 |
| *Bub1b*$^{AI}$ (3) | 150 | 8 (3) | 0 | 0 | 1 | 2 | 138 | 9 | 0 | 0 | 0 |
| *Bub1b*$^{AN}$ (5) | 250 | 18 (6)* | 0 | 0 | 2 | 15 | 204 | 23 | 6 | 0 | 0 |
| *Bub1b*$^{N}$ (3) | 150 | 6 (2) | 0 | 0 | 1 | 3 | 141 | 4 | 1 | 0 | 0 |

*$p < 0.05$.

**Source data 1.** Source file for MEF aneuploidy rate data.

FL-*Bub1b*. In addition to adenomas, Kras$^{G12D}$ produces a hyperplastic epithelium throughout the lung, which is prone to aneuploidization as evidenced by FISH for chromosomes 4 and 7 (***Figure 2B, D***) (***Baker et al., 2013***). Analysis of hyperplastic lung tissue from *Kras*$^{LA1}$ mice containing the FL-*Bub1b* transgene revealed that BubR1 overexpression has the ability to counteract Kras$^{G12D}$-mediated aneuploidization (***Baker et al., 2013***). Using FISH analysis, mice expressing the various *Bub1b* mutant transgenes revealed that tumor protection tightly correlated with an ability to counteract Kras$^{G12D}$-mediated aneuploidization (***Figure 2D***). Taken together, these data are consistent with the idea that BubR1 exerts its anti-neoplastic actions by preserving genomic integrity and suggest that only a subset of functional domains of BubR1 are necessary to afford protection against aneuploidization, which includes the N-terminal domain required for potent inhibition of APC/C$^{Cdc20}$ (***Malureanu et al., 2009***).

## Protection against aneuploidy and cancer correlates with increased BubR1 at kinetochores

To understand the molecular mechanism(s) underlying BubR1's protective qualities, we conducted an extensive comparative analysis between MEFs from transgenics expressing FL-BubR1 and our BubR1 mutants for their ability to engage pathways that safeguard against chromosome missegregation, including the SAC and the MT-KT attachment error correction machinery. First, we focused on BubR1-kinetochore localization, as this property has been shown to be important for its role in both error correction and SAC signaling (***Malureanu et al., 2009***). By immunostaining with antibodies

**Table 2.** *Bub1b* transgenic MEFs have normal missegregation rates. Live-cell imaging of chromosome segregation defects in primary H2B-RFP MEFs of indicated genotypes. $n \geq 3$ lines, $\geq 20$ cells per line. Data are mean ± s.e.m. WT, wild-type. FL, full-length. (See associated **Table 2—source data 1**).

| MEF genotype (n) | Mitotic cells inspected | Cells with segregation defects | Metaphases with misaligned chromosomes | Anaphases with lagging chromosomes | Anaphases with bridges chromosome |
| --- | --- | --- | --- | --- | --- |
| | | % (s.e.m) | % (s.e.m) | % (s.e.m) | % (s.e.m) |
| WT (3) | 94 | 16 (1) | 0 (0) | 6 (3) | 13 (1) |
| FL-*Bub1b* (3) | 105 | 17 (2) | 3 (3) | 3 (2) | 11 (2) |
| *Bub1b*$^{AI}$ (3) | 105 | 12 (1) | 1 (1) | 1 (1) | 10 (1) |
| *Bub1b*$^{AN}$ (3) | 101 | 19 (1) | 1 (1) | 7 (1) | 11 (2) |
| *Bub1b*$^{N}$ (3) | 95 | 17 (6) | 3 (2) | 2 (2) | 12 (6) |

**Source data 1.** Source file for missegregation assay data.

**Table 3.** *Bub1b* transgenic mice have normal rates of aneuploidy in vivo. Interphase FISH on specified tissues from mice of indicated genotypes. $n$ = 3 animals, 100 cells per tissue per animal. Data are mean ± s.d. WT, wild-type. FL, full-length. (See associated ***Table 3—source data 1***).

| Tissue Type | Genotype | Percentage of aneuploidy (s.d) | |
| --- | --- | --- | --- |
| | | Chrom 4 | Chrom 7 |
| Lung | WT | 1.3 (0.6) | 3.0 (0) |
| | FL-*Bub1b* | 2.3 (0.6) | 2.0 (1) |
| | *Bub1b*$^{\Delta I}$ | 2.0 (0) | 2.3 (0.6) |
| | *Bub1b*$^{\Delta N}$ | 3.0 (1) | 2.3 (0.6) |
| | *Bub1b*$^{N}$ | 2.3 (0.6) | 2.3 (0.6) |
| Heart | WT | 2.0 (1) | 1.7 (0.6) |
| | FL-*Bub1b* | 1.3 (0.6) | 2.0 (1) |
| | *Bub1b*$^{\Delta I}$ | 1.3 (0.6) | 2.0 (0) |
| | *Bub1b*$^{\Delta N}$ | 1.7 (1.2) | 2.0 (1) |
| | *Bub1b*$^{N}$ | 1.3 (0.6) | 1.7 (0.6) |
| Eye | WT | 2.0 (0) | 2.0 (1) |
| | FL-*Bub1b* | 2.0 (0) | 2.3 (0.6) |
| | *Bub1b*$^{\Delta I}$ | 1.7 (0.6) | 1.3 (0.6) |
| | *Bub1b*$^{\Delta N}$ | 2.0 (1) | 2.3 (0.6) |
| | *Bub1b*$^{N}$ | 1.7 (0.6) | 2.0 (0) |
| Kidney | WT | 2.0 (1) | 2.0 (1) |
| | FL-*Bub1b* | 2.3 (0.6) | 2.0 (0) |
| | *Bub1b*$^{\Delta I}$ | 2.0 (1) | 1.3 (0.6) |
| | *Bub1b*$^{\Delta N}$ | 2.7 (0.6) | 2.0 (0) |
| | *Bub1b*$^{N}$ | 2.0 (1) | 1.3 (0.6) |
| Spleen | WT | 3.3 (0.6) | 2.3 (1.2) |
| | FL-*Bub1b* | 3.0 (1) | 2.0 (1) |
| | *Bub1b*$^{\Delta I}$ | 2.0 (1) | 1.7 (0.6) |
| | *Bub1b*$^{\Delta N}$ | 3.0 (1) | 2.7 (0.6) |
| | *Bub1b*$^{N}$ | 2.7 (0.6) | 3.0 (0) |
| Skeletal muscle | WT | 2.3 (0.6) | 2.6 (0.6) |
| | FL-*Bub1b* | 2.7 (0.6) | 2.0 (0) |
| | *Bub1b*$^{\Delta I}$ | 2.0 (0) | 2.0 (1) |
| | *Bub1b*$^{\Delta N}$ | 2.3 (1.2) | 2.7 (0.6) |
| | *Bub1b*$^{N}$ | 2.0 (0) | 2.3 (0.6) |

**Source data 1.** Source file for tissue aneuploidy rate data.

directed against the BubR1 N-terminus, we found that both FL-*Bub1b* and *Bub1b*$^{\Delta I}$ prometaphases had markedly increased amounts of BubR1 compared to wild-type MEFs (***Figure 3A,B***). Due to antibody limitations, we were unable to distinguish the ratio of kinetochore-localized endogenous to transgenic BubR1 within the mutant MEFs, with the exception of *Bub1b*$^{N}$, in which only endogenous protein can be detected. This illuminated that endogenous BubR1 was displaced from the kinetochore, but we cannot rule out that the other mutants also had lower endogenous levels at the kinetochore, which is likely due to increased abundance of mutant protein within the cell. By staining with a Flag antibody that recognizes only the transgenic BubR1 protein, we determined that the relative expression of BubR1$^{\Delta I}$ was slightly higher than FL-BubR1, while BubR1$^{N}$ was equivalent to FL-BubR1

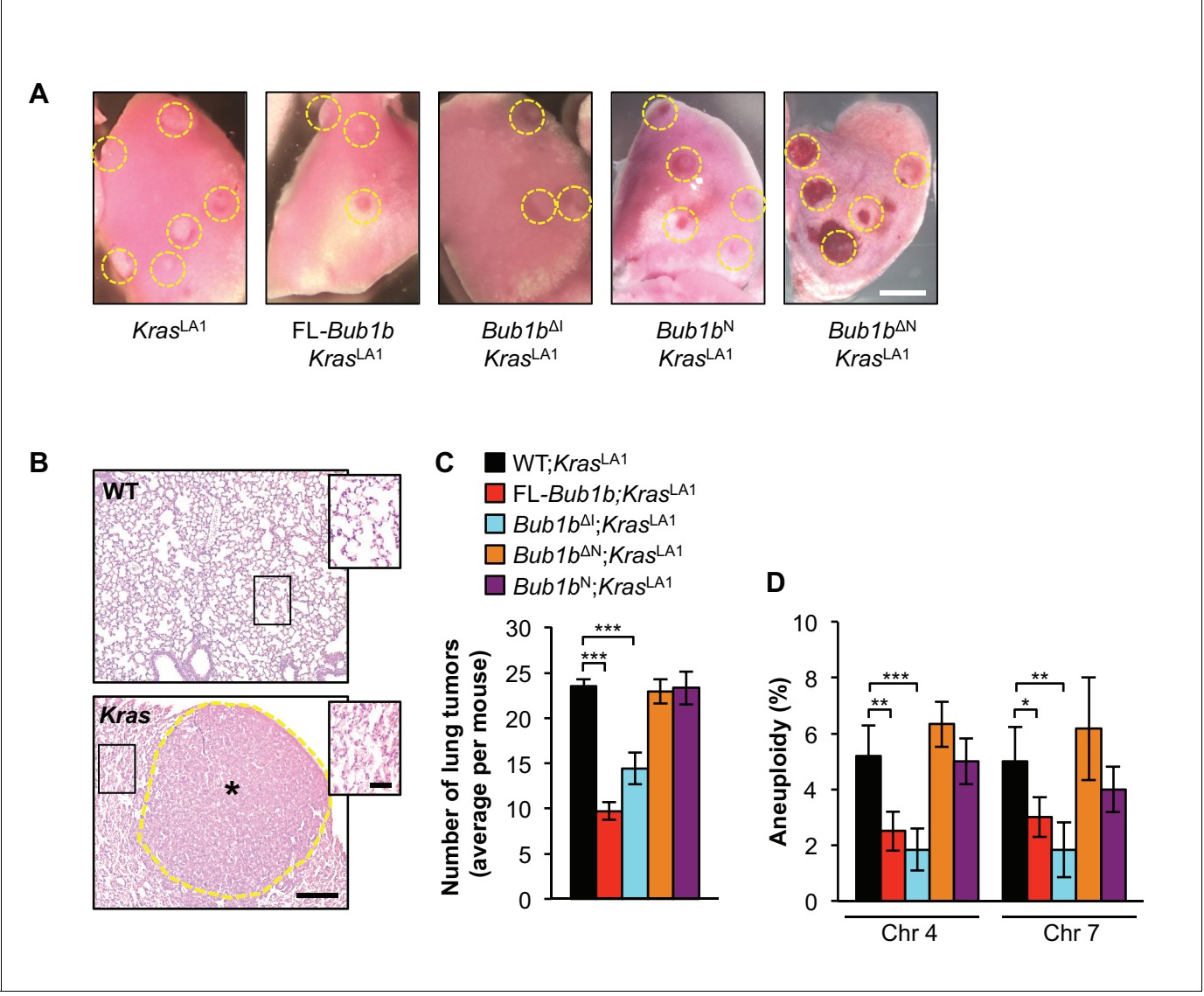

**Figure 2.** Select BubR1 domain overexpression protects against aneuploidy and cancer. (**A**) Lung lobes of $Kras^{LA1}$ mice and $Kras^{LA1}$ mice expressing various BubR1 transgenic proteins sacrificed at 6 weeks of age. Entire lungs were inspected using a dissection microscope to quantitate the number of lung tumors (adenomas) per mouse. (**B**) Hematoxylin-eosin stained lung sections of representative normal (WT) lung and a $Kras^{LA1}$ (Kras) hyperplastic tumor-bearing lung (the dashed line marks the adenoma boundary). Insets highlight normal and hyperplastic lung architecture. (**C**) Quantification of the number of lung tumors from mice shown in **A**. $n = 20$, except for full-length (FL)-$Bub1b$ where $n = 7$. Data are mean ± s.e.m. ***$p<0.001$. (**D**) Interphase FISH on the lungs of wild-type and $Kras^{LA1}$ with and without FL-BubR1 and mutant overexpression. $n = 5$, $\geq 100$ cells per animal. Data are mean ± s.d. *$p<0.05$, **$p<0.01$. ***$p<0.001$. Scale bars: **A**, 2 mm; **B**, 200 µm (main image) and 50 µm (insets). (See associated *Figure 2—source data 1*).

The following source data is available for figure 2:

**Source data 1.** Source file for tumor incidence and tissue aneuploidy rate data.

(*Figure 3C,D*). BubR1$^{\Delta N}$, which did not protect against cancer, failed to localize to kinetochores, consistent with a lack of the GLEBs motif (*Lampson and Kapoor, 2005*). Taken together, these data demonstrate a likely need for BubR1 to retain functionality, perhaps mediated through the N-terminal domain, at the kinetochore to prevent aneuploidy and tumorigenesis.

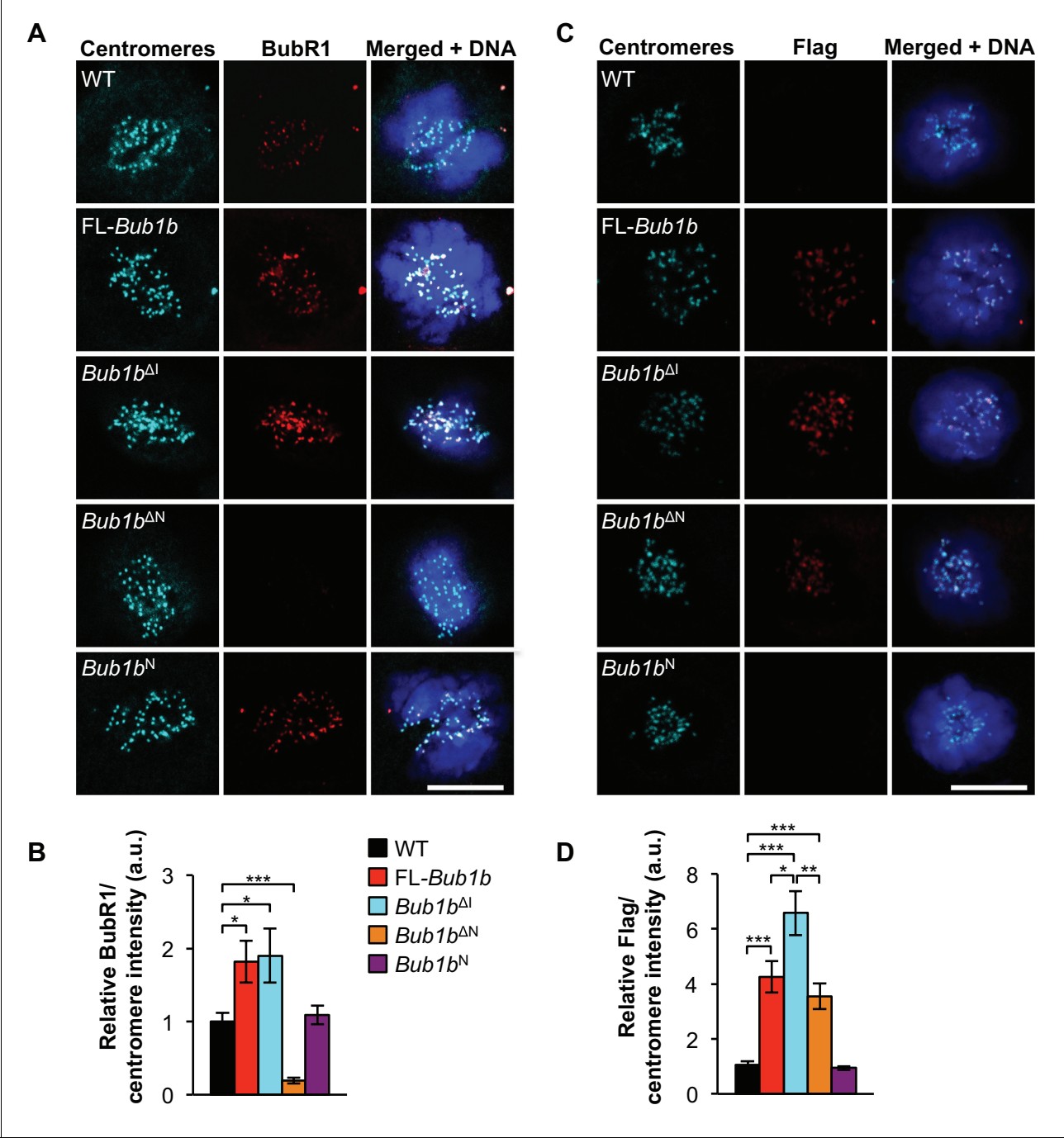

**Figure 3.** Increased BubR1 localization to kinetochore corresponds to phenotypic benefits. (**A**) MEFs of indicated genotypes were stained for BubR1 (red), centromeres (cyan), and DNA (blue). WT, wild-type. FL, full-length. (**B**) Quantification of immunostaining of BubR1 shown in **A**. Values were normalized to centromere stain and are relative to wild-type. $n = 3$ lines, $\geq 10$ cells per line. Data are mean ± s.d. *$p<0.05$, ***$p<0.001$. (**C**) Same as in **A** except with anti-Flag antibody to detect transgenic BubR1. (**D**) Quantification of immunostaining of Flag shown in **C**. Values were normalized to centromere stain and are relative to wild-type. Wild-type and $Bub1b^N$ represent background. $n = 3$ lines, $\geq 10$ cells per line. Data are mean ± s.d. *$p<0.05$, **$p<0.01$, ***$p<0.001$. Scale bar 10 μm. (See associated ***Figure 3—source data 1***).

The following source data is available for figure 3:

**Source data 1.** Source file for intensity of kinetochore-localized BubR1 and FLAG protein data.

# BubR1$^{\Delta I}$ extends metaphase and improves SAC sustainability

BubR1, Mad2, and Mps1 kinase set the speed limit for mitosis (*Meraldi et al., 2004*), and perturbations of these proteins accelerate mitotic timing and promote erroneous chromosome segregation (*Rodriguez-Bravo et al., 2014*). Therefore, we sought to determine the effect of FL-BubR1 and mutant BubR1 overexpression on mitotic timing by following MEFs from nuclear envelope breakdown (NEBD) to anaphase onset by time-lapse microscopy (*Figure 4A*). Because errors such as misalignments that may be caused by unattached kinetochores can trigger the SAC to delay mitotic progression, only cells that proceeded through mitosis without missegregation defects were

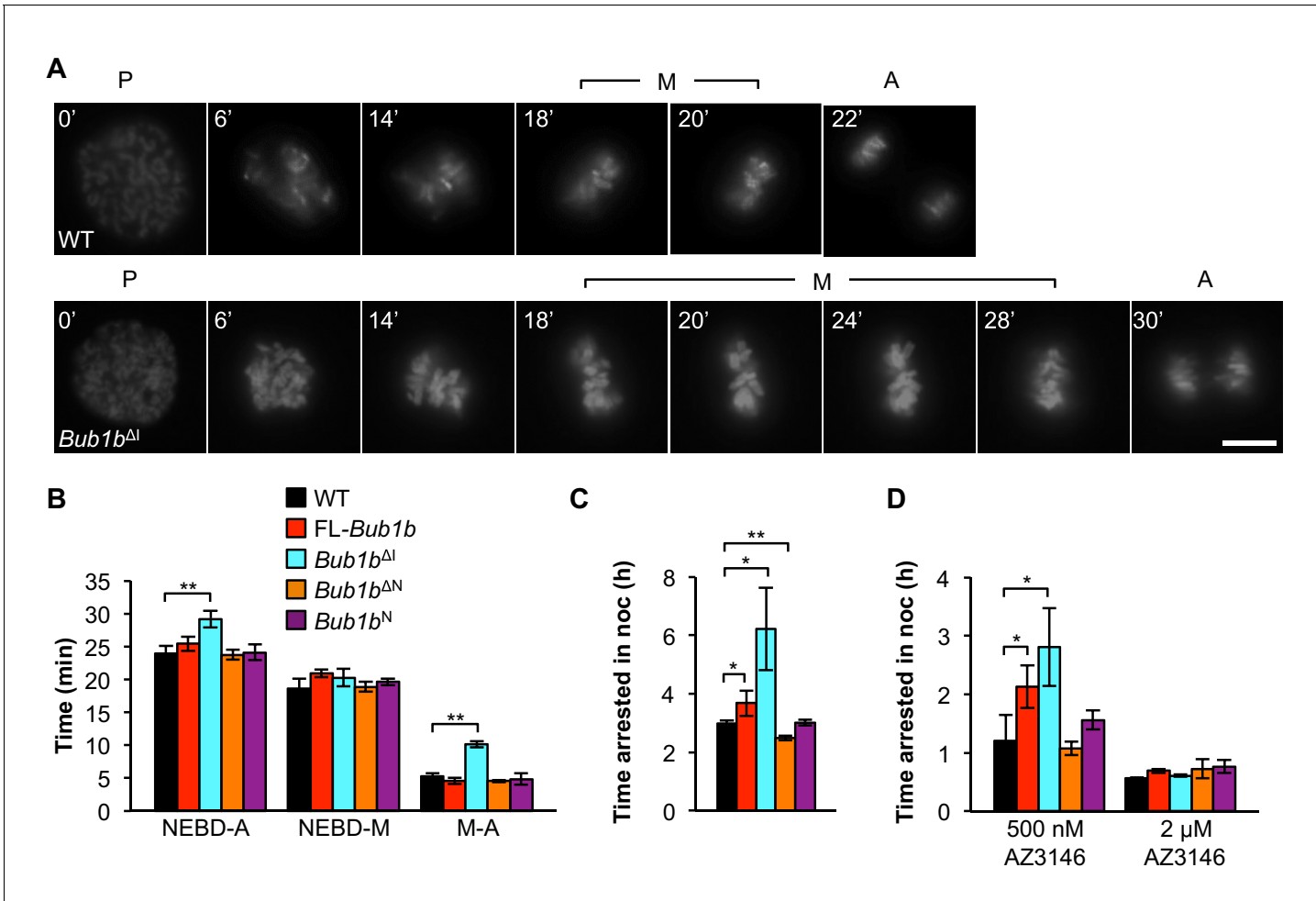

**Figure 4.** *Bub1b*$^{\Delta I}$ MEFs have an increased time in mitosis and duration of mitotic arrest. (**A**) Representative time-lapse images of live MEF cells of indicated genotypes progressing from prophase (t = 0) to anaphase (A). Time is indicated in min. WT, wild-type. P, prophase. M, metaphase. (**B**) Analysis of the time from nuclear envelope breakdown (NEBD) to anaphase onset in H2B-RFP MEFs of the indicated genotypes by live cell time-lapse imaging. *n* = 3 lines, ≥ 20 cells per line. Data are mean ± s.d. **p<0.01. FL, full-length. (**C**) In a nocodazole challenge, H2B-RFP MEFs of indicated genotypes were treated with 100 ng/ml of nocodazole (noc) and monitored by live cell time-lapse imaging. The point of time in which 50% of cells are arrested in mitosis is plotted. *n* ≥ 3 lines, ≥ 20 cells per line. Data are mean ± s.d. *p<0.05, **p<0.01. (**D**) H2B-RFP wild-type and mutant transgenic MEFs were treated concurrently with 100 ng/ml nocodazole and indicated concentrations of the Mps1 kinase inhibitor, AZ3146. The point of time in which 50% of cells are arrested in mitosis is plotted. *n* = 3 lines, ≥ 20 cells per line. Data are mean ± s.d. *p<0.05. Scale bar, 10 μm (See associated *Figure 4—source data 1*).

The following source data and figure supplement are available for figure 4:

**Source data 1.** Source file for mitotic timing and nocodazole arrest data.

**Figure supplement 1.** *Bub1b*$^{\Delta I}$ MEFs do not have persistent Mad2 signaling.

included. Overexpression of FL-BubR1 had no impact on mitotic timing and the same was true for BubR1$^N$ and BubR1$^{\Delta N}$ (*Figure 4B*). In contrast, *Bub1b*$^{\Delta I}$ MEFs spent significantly more time in mitosis (*Figure 4A,B*). The increase in mitotic timing was specifically attributed to the metaphase-to-ana-phase transition, where *Bub1b*$^{\Delta I}$ MEFs spent more than twice as long.

To verify that the extended time in mitosis was not due to unattached chromosomes, we per-formed an immunostaining for Mad2, which strongly localizes to unattached kinetochores (*Waters et al., 1998*). We found that while prometaphases of wild-type, FL-*Bub1b* and *Bub1b*$^{\Delta I}$ MEFs had many Mad2-positive kinetochores, there were very rare incidences of Mad2-positive kinet-ochores in metaphase (*Figure 4—figure supplement 1*). This indicated that the increased time of the metaphase-to-anaphase transition was delayed independent of unattached chromosomes or oth-erwise persistent Mad2 signaling. The extension of mitotic timing is a feature that is not shared with FL-overexpression alone, indicating it likely has a minimal contribution to tumor protection in our Kras model. However, it is particularly intriguing as a mechanism to prevent aneuploidy, as KT-MT attachment errors that are not detected by the SAC, namely merotelic attachments, may perhaps be given extra time to allow for the error correction machinery to prevent missegregation (*Cimini et al., 2001*; *Rodriguez-Bravo et al., 2014*).

Next, we examined whether overexpression of BubR1 and its variants impacted SAC sustainabil-ity. To do this, we added 100 ng/ml of the microtubule depolymerizing agent nocodazole and moni-tored the amount of time individual cells stayed arrested in mitosis. FL-BubR1 overexpression caused a slight but significant increase in duration of arrest, with transgenic cells arresting on aver-age for 3.7 hr compared to 3 hr for wild-type MEFs (*Figure 4C*). *Bub1b*$^N$ MEFs were unchanged from wild-type, while *Bub1b*$^{\Delta N}$ MEFs actually had a slight but significant reduction in arrest time. In contrast, *Bub1b*$^{\Delta I}$ MEFs showed a dramatic extension of checkpoint sustainability, with cells arresting for an average time of 6.2 hr. Thus, the two BubR1 alterations offering tumor protection, overex-pression of FL-BubR1 and BubR1$^{\Delta I}$, improve checkpoint sustainability although the latter does so much more robustly. In complementary experiments, we challenged the SAC by inhibiting Mps1, a kinase necessary both for the establishment and maintenance of the SAC (*Hewitt et al., 2010*; *Liu and Winey, 2012*). Treatment of MEFs concomitantly with 100 ng/ml nocodazole and a high concentration of the Mps1 inhibitor AZ3146 (2 μM) completely abolished SAC activity irrespective of the *Bub1b* transgene expressed (*Figure 4D*). At a four-fold lower inhibitor concentration, wild-type, *Bub1b*$^N$, *Bub1b*$^{\Delta N}$ MEFs were all capable of mounting a modest mitotic arrest. The relative exten-sion of mitotic arrest in FL-*Bub1b* and *Bub1b*$^{\Delta I}$ MEFs, however, was considerably higher, with *Bub1b*$^{\Delta I}$ MEFs reaching a similar level of SAC signaling in the presence of 0.5 μM AZ3146 as wild-type MEFs in the absence of inhibitor (*Figure 4D*). Thus, under normal SAC conditions and condi-tions where the SAC signaling is weakened, both BubR1$^{\Delta I}$ and FL-BubR1overexpression seem capa-ble of prolonging mitotic arrest, albeit to different degrees.

## BubR1$^{\Delta I}$ lowers the threshold for SAC activation

Because the internal Cdc20-binding domain of BubR1 has been implicated in both initiating and silencing the mitotic checkpoint (*Diaz-Martinez et al., 2015*; *Lischetti et al., 2014*), we hypothe-sized that FL-*Bub1b* and *Bub1b*$^{\Delta I}$ MEFs may have a lower threshold of checkpoint activation or a dif-ficulty silencing the checkpoint, or both. To examine whether FL-*Bub1b*and *Bub1b*$^{\Delta I}$ MEFs might have a lower threshold for SAC activation, we challenged them with low concentrations of nocoda-zole and monitored time to anaphase onset (*Figure 5A*). Based on the response of wild-type MEFs, we found that 20 ng/ml was the optimal dose to use in this experiment (*Figure 5A*). At this dose, however, the time *Bub1b*$^{\Delta I}$ MEFs took to go through mitosis increased by 60% compared to increases of ~25% in wild-type and FL-*Bub1b* MEFs, suggesting that these MEFs had a lower thresh-old for SAC activation.

Next, we sought to determine whether these MEFs also had difficulty in silencing the SAC. To test this we used a live-cell imaging-based approach in which we cultured MEFs in 100 ng/ml noco-dazole for 1.5 hr to activate the SAC and then monitored mitotically-arrested cells for time to mitotic exit following treatment with either vehicle (DMSO) or 2 μm AZ3146 as a stimulus for dissolving MCCs (*Figure 5B*). In this assay, neither FL-BubR1 nor BubR1$^{\Delta I}$ overexpression permitted an arrest longer than that observed in wild-type MEFs. Additionally, while BubR1-associated PP2A has been shown to be important for error correction in human cells (*Suijkerbuijk et al., 2012*), it is also impor-tant for silencing the SAC (*Espert et al., 2014*). As BubR1$^{\Delta I}$ lacks the KARD region implicated in

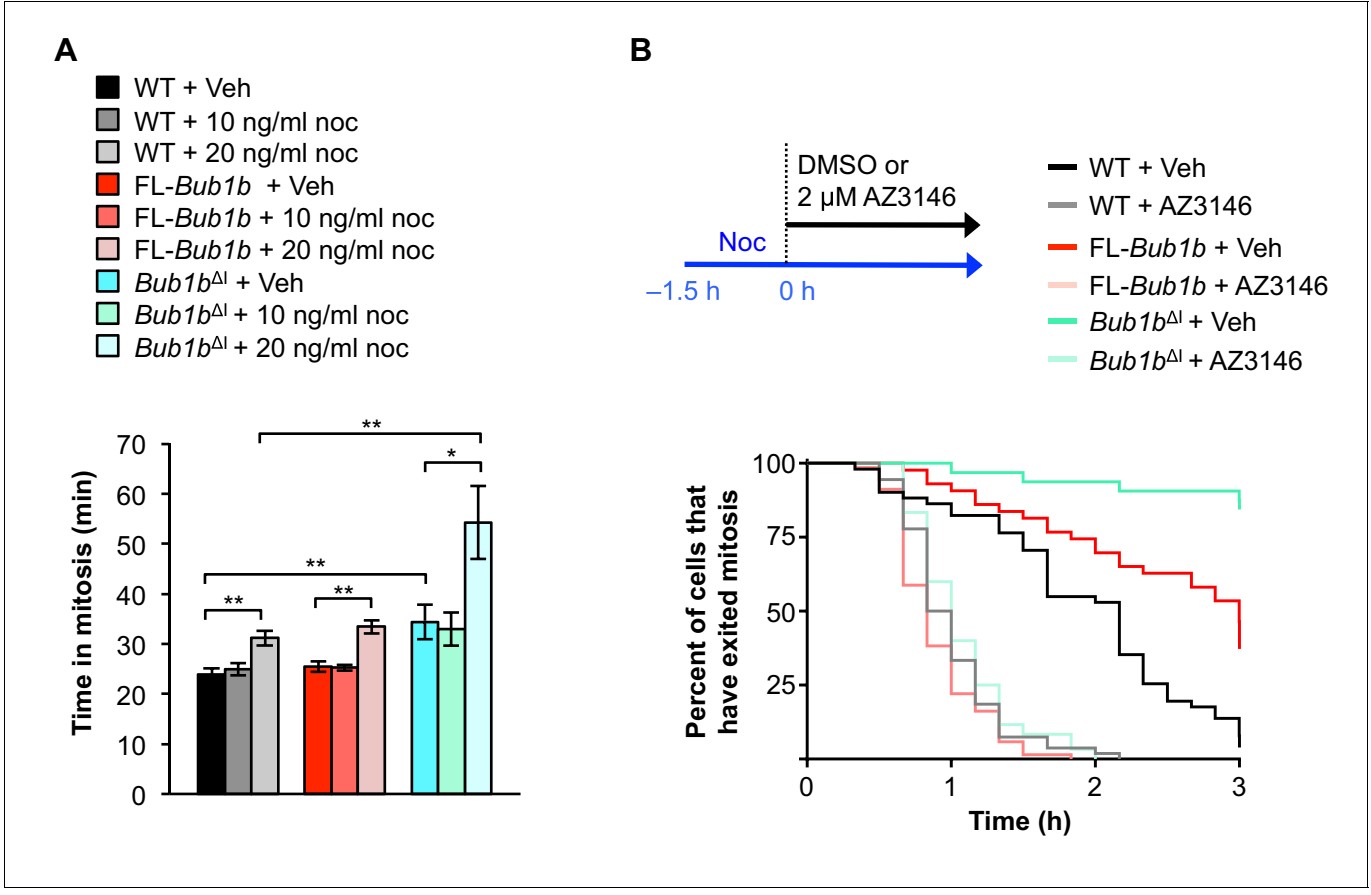

**Figure 5.** $Bub1b^{\Delta I}$ MEFs have a lower threshold to checkpoint activation. (**A**) Analysis of the time from NEBD to anaphase onset in H2B-RFP MEFs of the indicated genotypes treated with either DMSO vehicle (Veh) or indicated concentration of nocodoazole (Noc). $n$ = 3 lines, $\geq$ 20 cells per line. Data are mean ± s.d. *p<0.05, **p<0.01. WT, wild-type. FL, full-length. (**B**) (top) Strategy for analyzing the checkpoint silencing efficiency. MEFs of indicated genotypes were treated with 100 ng/ml nocodazole for 1.5 hr before addition of either DMSO vehicle (Veh) or 2 μM AZ3146, at which point cells were marked and monitored for time of escape (time point zero). (bottom) Analysis of duration of mitotic arrest from time point zero as outlined in (top). $n$ = 3 lines, $\geq$ 20 cells per line. (See associated *Figure 5—source data 1*).

The following source data and figure supplements are available for figure 5:

**Source data 1.** Source file for low-dose nocodazole challenge and SAC silencing data.

**Figure supplement 1.** PP2A localization is normal in $Bub1b^{\Delta I}$ MEFs.

**Figure supplement 1—source data 1.** Source file for intensity of kinetochore-localized PP2A protein data.

PP2A recruitment, its overexpression could potentially mislocalize PP2A and impede proper SAC silencing. We found PP2A localization to be normal in BubR1$^{\Delta I}$ overexpressing cells, suggesting this branch of signaling is not impacted (*Figure 5—figure supplement 1*), and further supporting that silencing of the SAC is not disrupted. Taken together, these results suggest that the threshold to engage the SAC is instead lowered by the $Bub1b^{\Delta I}$ transgene.

## The mitotic checkpoint complex composition is unique in $Bub1b^{\Delta I}$ MEFs

To explore the mechanism as to why $Bub1b^{\Delta I}$ and FL-$Bub1b$ MEFs both had more robust checkpoint signaling when challenged with nocodazole, we determined whether the amount of Cdc20 bound to BubR1 was increased in these cells. To this end, we treated wild-type, $Bub1b^{\Delta I}$ and FL-$Bub1b$ MEFs cells with nocodazole for 1 hr before harvesting them by mitotic shake-off. We found that overexpression of FL-BubR1 lead to increased interaction of BubR1 and Cdc20, as had been previously

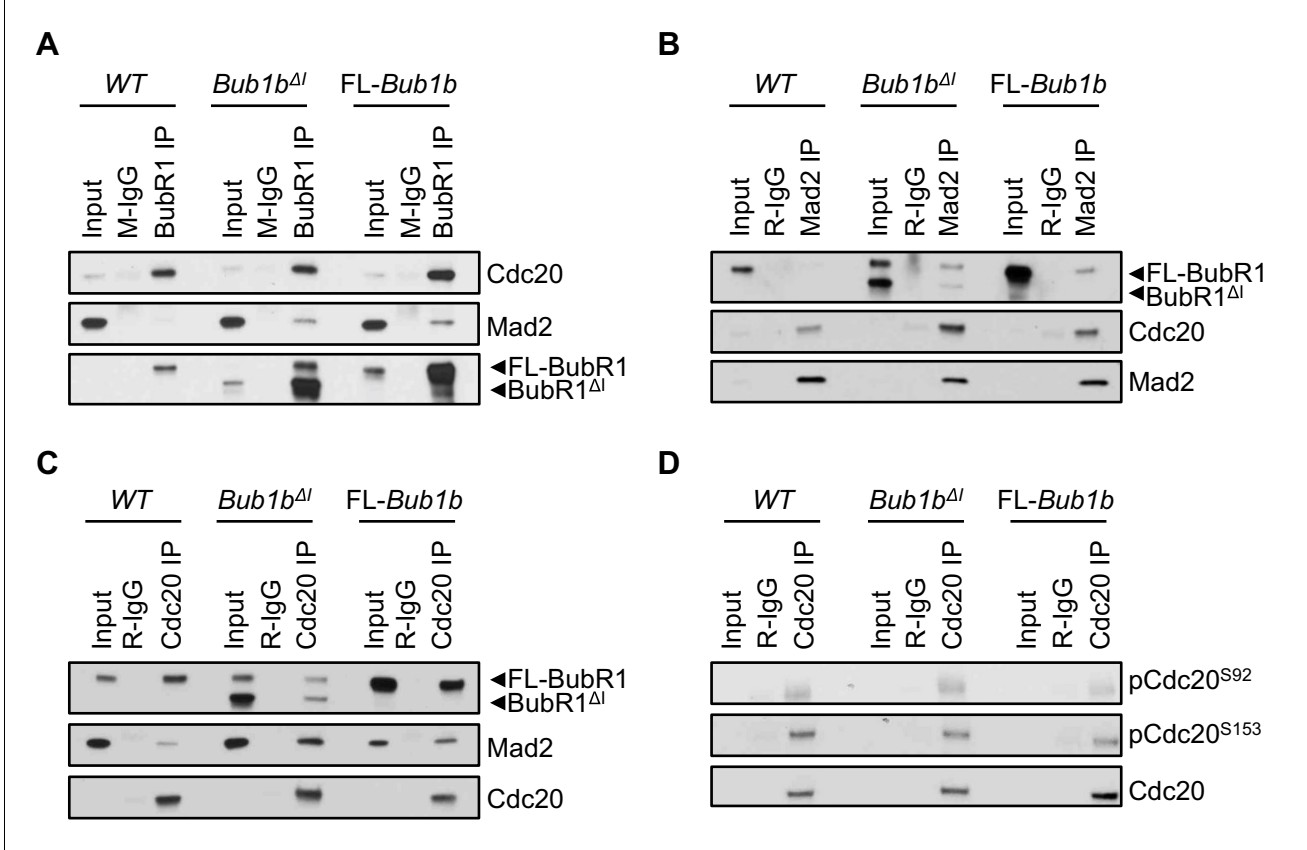

**Figure 6.** Composition of mitotic checkpoint complexes is unique in *Bub1b*^ΔI MEFs. (**A–D**) Immunoblots of mitotic wild-type (WT) and indicated mutant MEF extracts subjected to co-immunoprecipitation with the indicated antibodies. Each blot is a representative of at least 3 experiments. FL, full-length.

The following figure supplement is available for figure 6:

**Figure supplement 1.** Mitotic checkpoint components have a normal expression in *Bub1b* transgenic MEFs.

shown (*Figure 6A*) (*Baker et al., 2013*). This was confirmed by Cdc20 and Mad2 co-IPs and subsequent analysis of co-precipitating proteins (*Figure 6B,C*) that indicated there were an increased amount of complexes consisting of BubR1-Cdc20-Mad2, a potent APC/C inhibitor. However, we found that while the total amount of BubR1 (wild-type and mutant) immunoprecipitated in *Bub1b*^ΔI MEFs is increased substantially over normal MEFs, BubR1 lacking the internal Cdc20-binding domain failed to pull-down excess Cdc20 and vice versa (*Figure 6A,C*). This suggests that while BubR1^ΔI can likely bind to Cdc20, it is not a preferred partner. Surprisingly, the amount of Mad2 co-precipitated by Cdc20 was increased, indicating that a larger proportion of Cdc20 bound Mad2 than in wild-type MEFs, while retaining similar levels of BubR1-Cdc20-Mad2 complexes as wild-type. Immunoprecipitation of Mad2 and Cdc20 and analysis of co-precipitating MCC components confirmed this (*Figure 6B,C*). These unique MCC compositions did not result from changes to total levels of these proteins (*Figure 6—figure supplement 1*). Thus, FL-BubR1 overexpression alone results in the ability for cells to form more mitotic checkpoint complexes compared to wild-type MEFs, which can likely fortify the SAC signaling potential. *Bub1b*^ΔI, however, exerts its impacts on the MCC through a different mechanism. The increased abundance of Mad2-Cdc20 complexes, albeit a weaker inhibitor of APC/C than a full complement of the MCC, in addition to wild-type levels of MCC could represent a state in which the cells are poised to quickly activate the SAC. This is supported by our data in which *Bub1b*^ΔI show increased sensitivity to a weak SAC-inducing stimulus (*Figure 5A*).

In complimentary experiments, we sought to determine if an MCC-independent mechanism could also contribute to the extended SAC arrest of FL-*Bub1b* and *Bub1b*^ΔI MEFs. In addition to being

incorporated into the MCC, Cdc20 is subject to two regulatory phosphorylation events that disrupt its ability to activate the APC/C (*Jia et al., 2016*). We examined the phosphorylation status of two residues of Cdc20 implicated in mediating this inhibition, S153 and S92 by Bub1 kinase and Plk1 kinase respectively, in wild-type, FL-*Bub1b* and *Bub1b*$^{\Delta I}$ MEFs by Western blot (*Figure 6D*) (*Jia et al., 2016*). We found the levels of phosphorylation of both these residues to be equivalent to wild-type MEFs, suggesting this method of APC/C control is not hyperactive in our mutants.

## Protection from aneuploidy and tumorigenesis correlates with reinforced error correction

Next we investigated whether and how overexpressed FL-BubR1 and BubR1$^{\Delta I}$ contributed to high-fidelity chromosome segregation under mitotic duress by reinforcing the attachment error correction machinery. To do so, we used the motor protein Eg5 inhibitor monastrol to induce syntelic attachments, a malattachment type that presents as misaligned chromosomes and is resolved by the attachment error correction machinery (*Lampson et al., 2004*). Because error correction is highly efficient in wild-type MEFs, we challenged the machinery in our experimental system by limiting Aurora B kinase activity with 10 nM of the small molecule inhibitor AZD1152 (*Ricke et al., 2012*). FL-BubR1 and BubR1$^{\Delta I}$ both significantly blunted the increase in syntelic attachments caused by hypoactive Aurora B (*Figure 7*). In contrast, no such corrective effects were observed in the *Bub1b*$^{\Delta N}$ or *Bub1b*$^{N}$ mutants.

## Refined BubR1$^{\Delta I}$ mutants are capable of reinforcing error correction and SAC signaling

At the time of transgenic design, BubR1 residues 525–700 were defined as the internal Cdc20 domain (*Malureanu et al., 2009*). However, subsequent studies have revealed that this region includes at least three discrete functional units: the Phe-box, D-box2 and KARD, the first two of

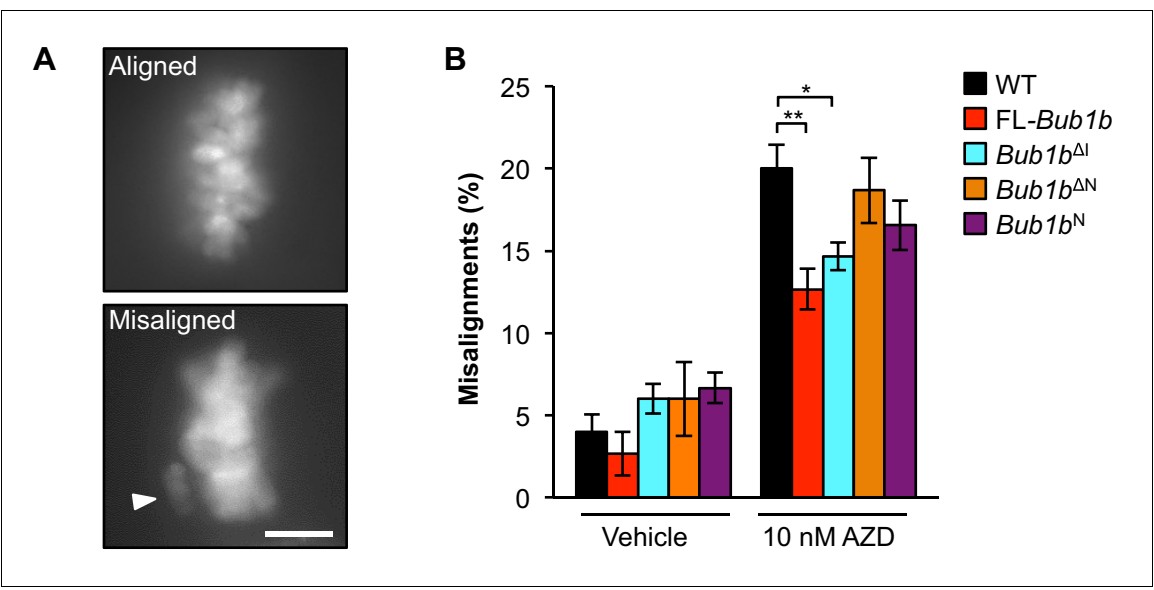

**Figure 7.** Overexpression of FL-BubR1 and BubR1$^{\Delta I}$improves error correction rates. (**A**) Representative images of MEFs with aligned or misaligned chromosomes after monastrol washout. White arrowhead depicts misaligned chromosome. (**B**) Analysis of chromosome misalignment in MEFs expressing the indicated *Bub1b* transgenes. MEFs were treated with 100 µM monastrol for 1 hr and then with monastrol and 10 µM MG132 for 1 hr and released for 90 min into MG132. Cells were treated with DMSO (Vehicle) or 10 nM AZD1152-HQPA (AZD) throughout the duration of the experiment. *n* = 6 lines (≥25 cells per line were analyzed). Data are mean ± s.d. *p<0.05, **p<0.01. Scale bar, 10 µm. WT, wild-type. (See associated *Figure 7—source data 1*).

The following source data is available for figure 7:

**Source data 1.** Source file for monastrol washout data.

which function as non-redundant Cdc20-binding motifs (*Diaz-Martinez et al., 2015*; *Lischetti et al., 2014*; *Suijkerbuijk et al., 2012*). This prompted us to study the extent to which BubR1 overexpression is able to preserve genomic stability when these functional units are deleted individually or in combination (*Figure 8*). These mutants were expressed in wild-type primary MEFs using a lentiviral expression system that allows for doxycycline inducible transgene expression. FL-BubR1 and BubR1$^{\Delta I}$ expressed in the same system were used as controls. We confirmed that each mutant was specifically and highly overexpressed in the presence of doxycycline (*Figure 8—figure supplement 1*).

Next, we examined the impact on mitotic timing by live cell imaging (*Figure 9A*). Overexpression of BubR1$^{\Delta I}$ provided a similar extension to the metaphase-to-anaphase transition as MEFs derived from *Bub1b*$^{\Delta I}$ transgenic mice. However, none of the other deletion constructs changed the duration of mitosis when overexpressed. An additional deletion mutant lacking the Phe box, D-box2 and the KARD (*Bub1b*$^{\Delta PheD\Delta KARD}$) was generated to test if combined deletion of all three motifs would mimic *Bub1b*$^{\Delta I}$ (*Figure 9—figure supplement 1*). Again, no extended mitotic timing was observed, implying that an unmapped domain within residues 525–700 regulates mitotic timing.

As expected, BubR1$^{\Delta I}$-expressing MEFs showed the most profound increase in the duration of nocodazole-mediated arrest, while FL-BubR1 overexpression caused a moderate, but significant increase (*Figure 4B*). Of our newly generated mutants, *Bub1b*$^{\Delta Phe}$, *Bub1b*$^{\Delta D}$ and *Bub1b*$^{\Delta KARD}$ behaved like overexpression of FL-BubR1, while *Bub1b*$^{\Delta PheD}$ phenocopied *Bub1b*$^{\Delta I}$. These findings indicate that none of the individual domains is required for SAC reinforcement by high levels of BubR1 and the combinatorial loss of both the Phe and D-box2 motifs is a requirement for robust

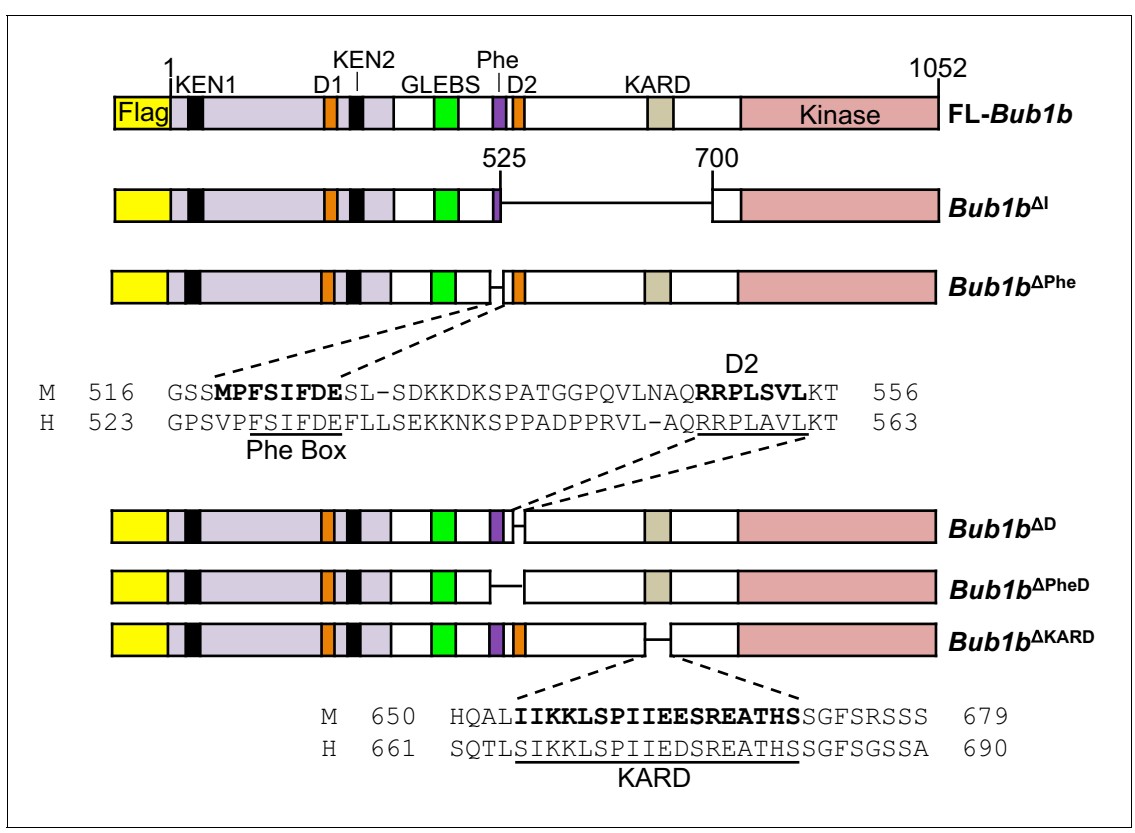

**Figure 8.** Schematics of pTripZ-Flag-*Bub1b* mutants. Schematics of the pTripZ-Flag-*Bub1b* constructs. D, destruction-(D-)box. GLEBs, GLEBs-binding motif. Phe, Phe box. KARD, kinetochore attachment regulatory domain. FL, full-length. Sequence alignment of the Phe box, D-box2, and KARD region of human and mouse BubR1. Residues characterized in human BubR1 are underlined, and homologous residues deleted in mouse *Bub1b* constructs are bold.

The following figure supplement is available for figure 8:

**Figure supplement 1.** Protein levels of *Bub1b* deletion constructs in wild-type MEFs.

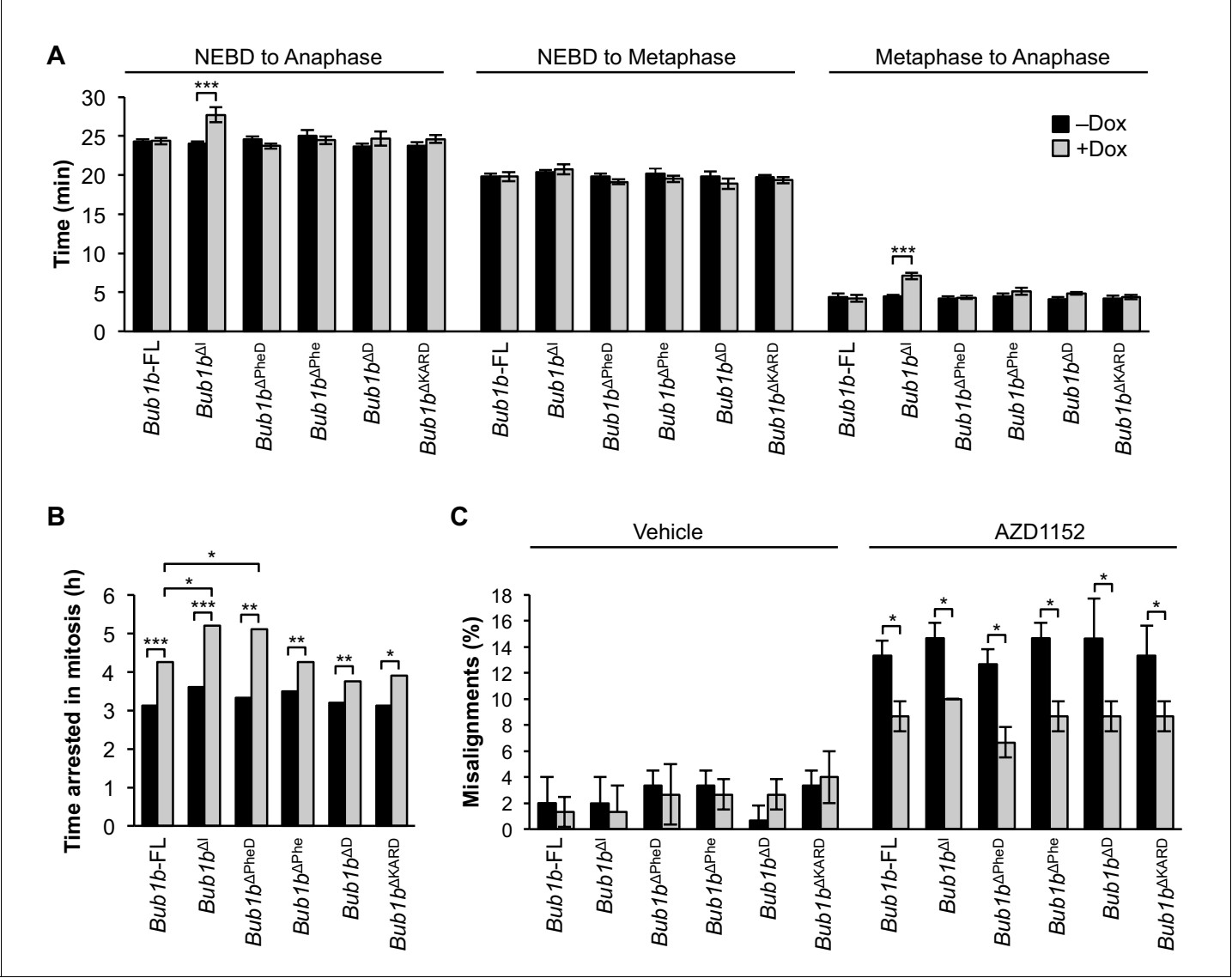

**Figure 9.** BubR1 deletion constructs extend nocodazole arrest and positively impact error attachment machinery. (**A**) Analysis of the time from nuclear envelope breakdown (NEBD) to anaphase onset in H2B-RFP wild-type MEFs infected with the indicated constructs with and without the addition of doxycycline (Dox) by live cell time-lapse imaging. $n$ = 1 line, ≥ 20 cells per treatment. Data are mean ± s.e.m. ***$p<0.001$. FL, full-length. (**B**) In a nocodazole challenge, H2B-RFP wild-type MEFs infected with the indicated constructs with and without the addition of Dox were treated with 100 ng/ml of nocodazole and monitored by live cell time-lapse imaging. The point of time in which 50% of cells are arrested in mitosis is plotted. $n$ = 1 line, ≥ 15 cells per treatment. *$p<0.05$, **$p<0.01$, ***$p<0.001$. (**C**) Analysis of chromosome misalignment in wild-type MEFs infected with the indicated constructs with and without addition of Dox. MEFs were treated with 100 µM monastrol for 1 hr and then with monastrol and 10 µM MG132 for 1 hr and released for 90 min into MG132. Cells were treated with DMSO (Vehicle) or 50 nM AZD1152-HQPA (AZD1152) throughout the duration of the experiment. $n$ = 3 lines, 50 cells per line per treatment. Data are mean ± s.d. *$p<0.05$. (See associated *Figure 9—source data 1*).

The following source data and figure supplements are available for figure 9:

**Source data 1.** Source file for mitotic timing, nocodazole challenge and monastrol washout data.
**Figure supplement 1.** Combined loss of Phe, D-box2 and KARD does not impact mitotic timing.
**Figure supplement 1—source data 1.** Source file for mitotic timing data.

checkpoint sustainability. When examined for the ability to improve microtubule-kinetochore attachment error correction in monastrol washout assays with and without Aurora B inhibition, all mutants did so to a similar extent as FL-BubR1 (*Figure 9C*), indicating that this feature of BubR1 overexpression is not impacted by any functional units in the central portion of BubR1. Altogether, our studies using refined BubR1$^{\Delta l}$ mutants indicate that individual domains within the 525–700 region are not required for SAC and error correction reinforcement by BubR1 overexpression, and that maximal SAC reinforcement is achieved when both internal Cdc20-binding motifs are absent.

## Cells expressing oncogenic Kras are prone to microtubule-kinetochore malattachment

Overexpression of FL-BubR1 and BubR1$^{\Delta l}$ reduces tumor burden and aneuploidization in *Kras*$^{La1}$ mutant mice. To better understand the underlying mechanisms, we determined the type of mitotic errors caused by oncogenic Ras in wild-type MEFs. We found that ectopic expression of Kras$^{G12V}$ had no impact on timing of mitosis and SAC signaling (*Figure 10A–C*). On the other hand, in monastrol washout assays, Kras$^{G12V}$-expressing MEFs produced significantly higher rates of misaligned chromosomes over empty vector alone (*Figure 10D*). In complementary experiments in which we monitored chromosome segregation errors by live cell imaging, Kras$^{G12V}$-expressing MEFs showed a remarkable increase in misaligned chromosomes (*Figure 10E*). Collectively, these data suggest that oncogenic Kras cells may be particularly prone to syntelic attachments.

## Discussion

BubR1 is different from other mitotic regulators in that supranormal expression improves numerical chromosomal integrity. Although it will be impractical to overexpress BubR1 for therapeutic purposes, increased insight into the molecular mechanisms underlying the positive effects of BubR1 overexpression might create entry points for development of novel anti-cancer treatments based on small molecules that complement current therapies. As a first step, we focused on the roles of BubR1's Cdc20-binding domains. Here we show that overexpression of the BubR1 N-terminal region is necessary, but not sufficient to prevent Kras-mediated aneuploidy and tumorigenesis, implying that reinforcing pseudo-substrate inhibition of Cdc20-bound APC/C by BubR1 is a requirement but entirely ineffective in isolation. In contrast to the N-terminal Cdc20-binding domain, a region spanning residues 525–700 that includes the elements of the internal Cdc20-binding domain and KARD, is dispensable for the aneuploidy and tumor suppressing effects. Overexpression of BubR1 lacking this region reinforced genomic stability and cancer protection in mice, despite dramatically altering metaphase duration of regularly dividing, unchallenged, MEF cells. We find that key shared characteristics of overexpressed FL-BubR1 and BubR1$^{\Delta l}$ include increased kinetochore localization, increased error correction ability and a more robust SAC, though at differing magnitudes. BubR1$^{\Delta l}$ overexpression was most extreme in altering the SAC in that it appeared to lower the threshold for checkpoint activation and maintenance. By analyzing the mitotic phenotypes of more refined deletion mutants, we determined that the profound length of nocodazole-induced mitotic arrest in this mutant was likely provided by a combined loss of the Phe box and D-box2, which is in alignment to a previous report showing that this region normally serves to shorten mitotic arrest times (*Diaz-Martinez et al., 2015*). However, whether the increased duration of mitotic arrest is actively participating in attenuation of Kras-mediated tumorigenesis in either FL-*Bub1b* or *Bub1b*$^{\Delta l}$ *Kras*$^{La1}$ mice is unclear, as oncogenic Kras alone did not negatively impact SAC signaling in MEFs.

Another unique attribute of BubR1$^{\Delta l}$ overexpression was its impact on normal mitotic timing. In pursuing this phenotype, we found that while overexpression of BubR1$^{\Delta PheD}$ recapitulated the robust nocodazole arrest seen in *Bub1b* MEFs, it did not reproduce the extension of the metaphase-to-anaphase transition. We further explored this with a mutant BubR1 lacking a combination of the Phe box, D-box2 and KARD (*Bub1b*$^{\Delta Phe\Delta KARD}$), but also did not see changes to mitotic timing. This suggests that the loss of a region within BubR1 between residues 525 and 700, either alone or in combination with the aforementioned domains, might be responsible for influencing the duration of mitosis. Further expansion of this notion could assign new functions related to timing to previously unmapped regions of BubR1.

Our studies in MEFs which had been infected with oncogenic Kras illuminated that while they had normal mitotic timing and SAC signaling, they had challenges with proper error correction

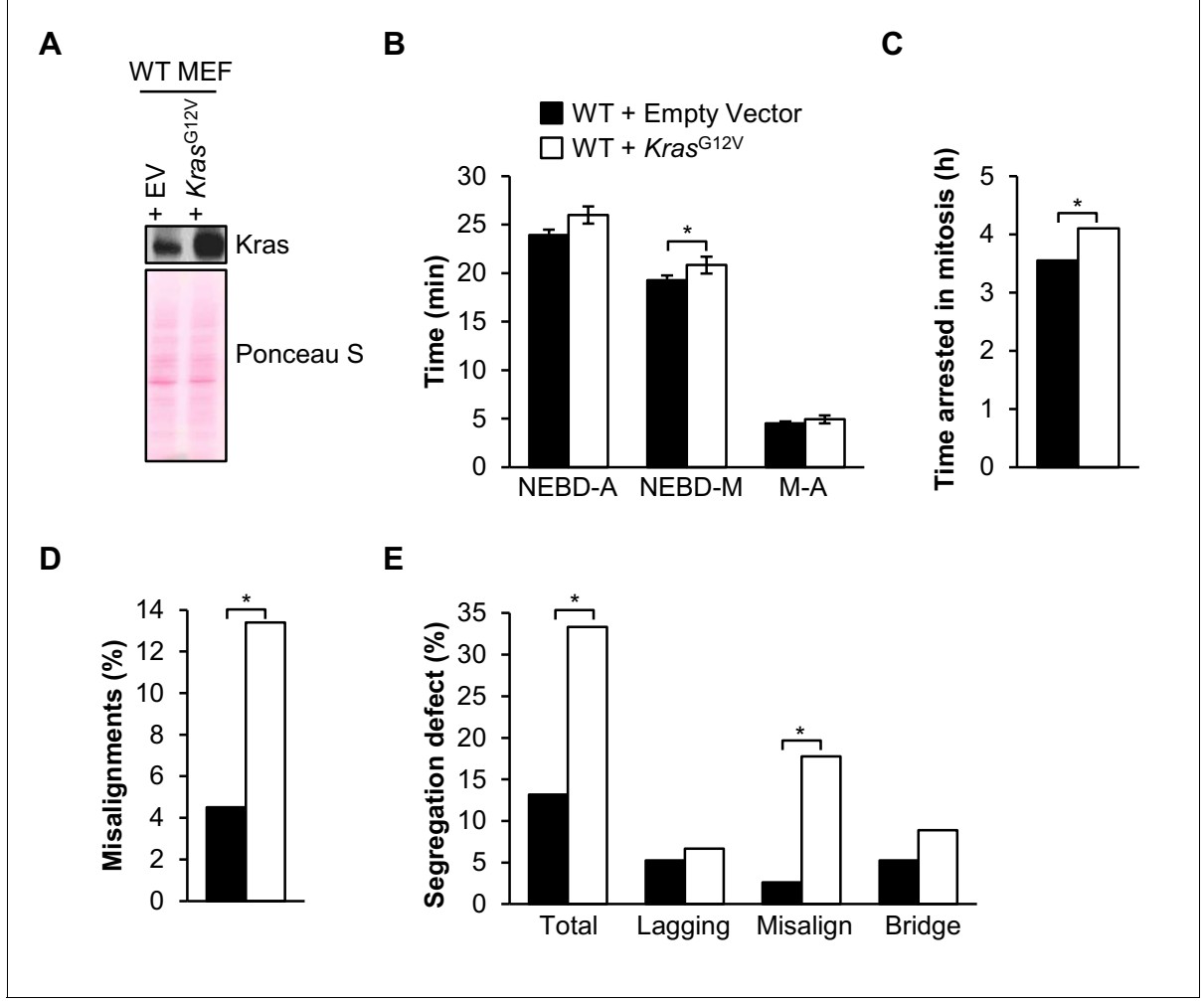

**Figure 10.** Oncogenic Kras increases microtubule-kinetochore malattachment. (**A**) Western blot of wild-type (WT) MEFs infected with pBABE-Puro-KRas (G12V) or empty vector (EV). Blot was probed with the indicated antibody. Ponceau S was used to normalize loading. (**B**) Analysis of the time from nuclear envelope breakdown (NEBD) to anaphase (A) onset in H2B-RFP wild-type MEFs infected with Kras$^{G12V}$ or EV by live cell time-lapse imaging. $n$ = 1 line, $\geq$ 19 cells. Data are mean ± s.d. *p<0.05. M, metaphase (**C**) In a nocodazole challenge, H2B-RFP wild-type MEFs infected with Kras$^{G12V}$ or EV were treated with 100 ng/ml of nocodazole and monitored by live cell time-lapse imaging. The point of time in which 50% of cells are arrested in mitosis is plotted. $n$ = 1 line, $\geq$ 20 cells. *p<0.05. (**D**) Analysis of chromosome misalignments in wild-type MEFs infected with Kras$^{G12V}$ or EV. MEFs were treated with 100 μM monastrol for 1 hr and then with monastrol and 10 μM MG132 for 1 hr and released for 90 min into MG132. $n$ = 1 line, ≈200 cells. *p<0.05. (**E**) Live-cell imaging of chromosome segregation defects in primary H2B-RFP wild-type MEFs infected with Kras or EV. $n$ = 1 line, ≈40 cells. *p<0.05. (See associated *Figure 10—source data 1*).

The following source data is available for figure 10:

**Source data 1.** Source file for mitotic timing, nocodazole challenge, missegregation assay and monastrol washout data.

machinery. Taken altogether, these data suggest that specific manipulations of BubR1 and subsequent protection against aneuploidy can occur through distinct complimentary mechanisms that result in tumor protection. In FL-*Bub1b* and *Bub1b*$^{\Delta I}$, parameters that positively influence genomic stability such as a robust SAC and improved error correction correlate with loss of Kras-mediated aneuploidy and tumorigenesis. Therefore, it is tempting to speculate that the strongest mechanism preventing aneuploidization in our Kras system might be a strengthening of the error correction machinery. Given the heterogeneous nature of lung tumors compared to MEFs, however, it is possible that distinct challenges and defects can arise in Kras lung tumors that are not overtly evident in MEFs.

Lung tissues with activated oncogenic Kras proceed through several morphological stages, including regions of mild hyperplasia/dysplasia that have increased aneuploidy relative to normal lung tissue, to small alveolar adenomas and finally culminating into overt carcinoma (*Baker et al., 2013*; *Johnson et al., 2001*). We proposed two distinct but non-mutually exclusive mechanisms of tumor protection. In the first mechanism, the impact of a hypersensitive SAC combined with an ability to promote proper attachments and prevent misalignments by FL-BubR1 and BubR1$^{\Delta I}$ simply prevents genetic heterogeneity that facilitates cancer progression. Thus, while hyperplasia is still a feature of the lungs, there is a block to full neoplastic transformation because necessary losses of tumor suppressors and gains of cancer promoting genes are prevented.

The second scenario is that survival of pre-neoplastic cells within the hyperplastic tissue is impacted by BubR1 overexpression, such that unstable cells have an increased propensity to die. The competing-networks model proposes that two independent yet competing cell fates oppose each other during mitotic arrest: death by caspase-mediated apoptosis or mitotic slippage resulting from persistent cyclin B1 degradation due to incomplete SAC inhibition (*Brito and Rieder, 2006*; *Gascoigne and Taylor, 2008*; *Topham et al., 2015*). Cells with prolonged mitotic arrest have a greater chance and more time to accumulate death signals (*Gascoigne and Taylor, 2008*). Both mutants have an impact on one arm of the competing network branches, as evidenced by their increase in arrest time in mitosis, though BubR1$^{\Delta I}$ appears to have a more significant contribution. This could shift a given cell population towards death rather than survival. However, this hypothesis remains to be rigorously tested in our model especially since oncogenic Kras-infected MEF cells do not show evidence of a weakened mitotic checkpoint. We argue that taking advantage of an extended arrest independent of whether or not the machinery functions normally could still be used to promote cancer cell death, and that BubR1 overexpression would be an entry point to such therapies. Along these lines, microtubules have been targeted in anti-cancer chemotherapy with much success (*Dumontet and Jordan, 2010*). Furthermore, the use of Wee/Chk1 inhibitors which cause cells to bypass the G2 checkpoint, force reliance on the SAC (*Mc Gee, 2015*). This could be exploited and we would predict an increased susceptibility to microtubule poisons and cytotoxicity, in particular with a system mimicking BubR1$^{\Delta I}$ or BubR1$^{\Delta PheD}$ overexpression.

Previously we reported a non-significant increase in the mitotic arrest time of MEFs overexpressing FL-BubR1 when challenged with nocodazole relative to wild-type (*Baker et al., 2013*). Here, we conclude, based on more in depth and sophisticated studies to test this aspect of SAC signaling, that there is a modest but significant increase in SAC potency in these cells (*Baker et al., 2013*). Furthermore, overexpression of BubR1$^{\Delta Phe}$, BubR1$^{\Delta D}$, and BubR1$^{\Delta KARD}$ deletion constructs in wild-type MEFs also provided a slight but significant increase in mitotic arrest time compared to controls, indicating that they were analogous to FL-BubR1 overexpression alone. It is important to note that the data provided by the extensive analysis of *Bub1b$^{\Delta I}$* and *Bub1b$^{\Delta KARD}$* MEFs are in disagreement with previous work in human cells, where the KARD was proposed to provide the MT-KT attachment function of BubR1 (*Suijkerbuijk et al., 2012*). In these studies, introducing exogenous BubR1 with a mutated KARD into systems depleted of endogenous BubR1 have decreased PP2A kinetochore localization, and subsequent chromosome alignment defects (*Suijkerbuijk et al., 2012*). In *Bub1b$^{\Delta I}$* MEFs, however, we do not see increased alignment errors or decreased PP2A kinetochore localization (*Table 2*, *Figure 5—figure supplement 1*) and both overexpressed BubR1$^{\Delta I}$ and BubR1$^{\Delta KARD}$ actually have increased error correction capabilities (*Figure 7B*, *Figure 9C*). We emphasize that unlike the studies by Suijkerbuijk and colleagues, our transgene and deletion construct are on the background of a full complement of endogenous BubR1, which may still provide adequate kinetochore docking of PP2A. Furthermore, our data are also in alignment with our previous work in which BubR1$^{\Delta I}$ expressed on a BubR1$^{-/-}$ background does not show overt increases in misalignments (*Malureanu et al., 2009*). This could represent species-specific differences in the reliance of the KARD for attachment, or on the dependency of BubR1 to recruit PP2A to the kinetochore.

Overexpression of BubR1$^{\Delta N}$ or its complimentary fragment BubR1$^{N}$ does not recapitulate benefits observed with FL-BubR1 and BubR1$^{\Delta I}$ overexpression. Instead, BubR1$^{\Delta N}$ may be imposing detrimental effects on the cells, as its overexpression results in a slight increase in aneuploidy and a decreased ability to sustain a checkpoint arrest in MEFs, though in vivo aneuploidy rates do not change. Interestingly, however, while the BubR1 N-terminus appears necessary for physiological benefits, it alone did not exert an anti-tumor effect. FL-*Bub1b* and *Bub1b$^{\Delta I}$* MEFs have increased BubR1 expression at kinetochores (*Figure 2B*), corresponding to phenotypic benefits. BubR1$^{N}$

cannot localize to kinetochores, in addition to lacking several functional domains such as the putative kinase/pseudokinase domain, the internal Cdc20-binding domain, and KARD. As we have determined from the studies herein that the internal binding domain and KARD are dispensable, kinetochore localization and subsequent action of BubR1 there might be key to beneficial phenotypes, in particular, error attachment (*Lampson and Kapoor, 2005*). This is in agreement with our previous study in which we found BubR1 that is unable to localize to kinetochores due to disruptions of the Bub3-binding domain cannot fully rescue spindle assembly checkpoint or mediate complete corrective effects on misalignment in cells depleted of endogenous BubR1 (*Malureanu et al., 2009*). The use of transgenic strains in which BubR1$^N$ is artificially tethered to kinetochores and evaluating the impact on SAC and error correction signaling could shed light on this in future studies (*Maldonado and Kapoor, 2011*). Other transgenic models could parse out the specific contribution of kinetochore-localized overexpression to tumor prevention with a domain mutant that does not permit BubR1 to localize to the kinetochore, as well as overexpression of a kinase-dead BubR1.

Whether or not aneuploidy causes cancer or is simply a feature is a longstanding question (*Giam and Rancati, 2015*). Here, we offer a unique perspective by showing the reinforcement of genomic stability through several complimentary mechanisms, with an emphasis on error correction machinery, attenuates tumors in a Kras mouse model of lung cancer. Tumor cells often have compromised DNA repair pathways, and are therefore sensitized to chemotherapies such as topoisomerase inhibitors and alkylating agents that promote cellular damage (*Calderón-Montaño et al., 2014*). There are few, if any, cancer treatments that revolve around promoting chromosomal stability and reinforcing known checkpoint pathways. Thus, by forcing reliance on the SAC and death in mitosis by damaging or bypassing other cell-cycle checkpoints, a more potent anti-tumor therapy could be designed.

## Materials and methods

### Mouse strains and husbandry

All protocols were reviewed and approved by the Mayo Clinic Institutional Animal Care and Use Committee. Mice used in these studies were housed in a pathogen-free barrier and are maintained on a mixed 129SV/E × C57BL/6 genetic background. Full-length Flag-m*Bub1b*transgenic mice have been described previously (*Baker et al., 2013*), and the generation of Flag-m*Bub1b*$^N$, Flag-m*Bub1b*$^{\Delta I}$, and Flag-m*Bub1b*$^{\Delta N}$ was performed using a similar strategy. The development of constructs for these fragments of BubR1 has been described in detail (*Malureanu et al., 2009*). Tumor studies of *Kras*$^{LA1}$ mice were performed as previously described (*Baker et al., 2013*). *Kras*$^{LA1}$ mice were obtained from the MMHCC (NCI Frederick) (*Johnson et al., 2001*). Mice were sacrificed at 6 weeks of age and surface adenomas were counted using a dissection microscope. Formalin-fixed, paraffin-embedded lung samples were stained for histological analysis using routine haematoxylin and eosin staining.

### Creation of inducible BubR1 deletion mutants

pTripz-Flag-FL-*Bub1b* was created from pTripz-PKG-puro-loxp (GE Dharmacon, Layfayette, CO). The loxp sites were removed and a multiple cloning site (MCS) was inserted. Flag-FL-*Bub1b* was removed from pMSCV-IRES-GFP (*Malureanu et al., 2009*) and cloned into the MluI site of the MCS. Deletion constructs were generated using the Q5 Site-Directed Mutagenesis Kit (New England Biolabs, Ipswich, MA; #E0554S) following manufacturer's instructions from template pTripz-Flag-FL-*Bub1b*. The following primers were used to create the following deletion constructs: pTripZ-Flag-*Bub1b*ΔI, Fwd 3'- GACGGGGCAGAAAATGCT-5', Rev 3'- AAAAATGGAGAAAGGCATACTG-5'; pTripZ-Flag- *Bub1b*ΔKARD, Fwd 3'- TCTGGCTTCTCCAGGTCTT-5', Rev 3'- GAGGGCCTGGTGATGAAC-5'; pTripZ-Flag-*Bub1b*ΔPhe, Fwd 3'- TCTCTTTCAGACAAAAAGGAC-5'; Rev 3'- ACTGGAACCTTTAGAATCAG-5'; pTripZ-Flag-*Bub1b*ΔD, Fwd 3'- AAAACTACAGAAGTGGGC-5', Rev 3'- CTGGGCATTGAGAACCTG-5'; pTripZ-Flag-*Bub1b*ΔPheD, Fwd 3'- AAAACTACAGAAGTGGGC-5', Rev 3'- CTGGAACCTTTAGAATCAG-5'. We used a similar approach to create pTripZ-Flag-*Bub1b*ΔPheDΔKARD using pTripZ-Flag-*Bub1b*ΔPheD forward and reverse primers and pTripZ-Flag-*Bub1b*ΔKARD as a template. Cloned plasmids were transfected into HEK-293T cells using the Trans-Lentiviral shRNA packaging kit with calcium phosphate (GE Dharmacon; #TLP5912) and virus was

harvested 48 hr post-transfection. Primary wild-type p3 MEFs were infected twice every 8 hr and selected with 2 µg/ml puromycin (InvivoGen, San Diego, CA) 48 hr post initial infection. At this time, 1 µg/ml doxycycline (Clontech, Mountain View, CA) was added and 48 hr later, cells were processed for western blotting or monastrol washout, or infected with H2B-RFP for live cell imaging.

### In vitro Kras studies

Wild-type MEFs were infected with pBABE-Puro-KRas(G12V) (Addgene plasmid #46746) or empty vector (Addgene plasmid #1764) and selected with 2 µg/ml puromycin 48 hr post-infection with cells for live-cell imaging being infected with H2B-RFP at this time. After 48 hr of selection, cells were processed for western blotting, monastrol washout, or live cell imaging.

### Generation and culture of MEFs

Wild-type and *Bub1b* transgenic MEFs were generated and cultured as described previously (*Baker et al., 2004*). At least three independently generated MEF lines per genotype were used unless otherwise stated. Asynchronous and mitotic shake-off MEF lysates were created as described previously (*Baker et al., 2013*).

### Western blotting and co-immunoprecipitation

Western blot analysis was performed as previously described (*van Ree et al., 2010*). Lung tissue lysates were prepared as previously described (*Baker et al., 2013*). Briefly, the lung tissue was snap-frozen in liquid nitrogen and then ground into powder with a mortar and pestle. Ten milligrams of the powder was resuspended in 100 µl of PBS, boiled for 10 min at 100°C after the addition of 100 µl Laemmli lysis buffer (Bio-Rad, Hercules, CA), and loaded into Tris-HCl polyacrylamide gels (Bio-Rad). Primary antibodies used were mouse anti-BubR1 (BD Transduction, San Jose, CA; 612503, 1:1,000), rabbit anti-mouse BubR1 (aa382-420) ([*Baker et al., 2004*]; 1:1000), rabbit anti-human BubR1 (aa1-350) ([*Baker et al., 2004*]; 1:1000), rabbit anti-Flag (Sigma-Aldrich, St. Louis, MO; F7425, 1:1000), rabbit anti-Flag (Cell Signaling, Danvers, MA; 2368S, 1:1000), rabbit anti-Cdc20 (Santa Cruz, Dallas, TX; sc-8358, 1:1000), mouse anti-Kras (Santa Cruz; sc-30, 1:1000) and rabbit anti-pCdc20$^{S92}$ and rabbit anti-pCdc20$^{S153}$ (generous gifts from Hongtau Yu). All antibodies were detected with secondary HRP-conjugated goat anti-mouse or anti-rabbit antibodies (Jackson Immunoresearch, West Grove, PA; 1:10,000). Ponceau S staining (1% glacial acetic acid, 1.1 g/ml Ponceau S [Sigma-Aldrich]) served as a loading control for blots. All western data are representative for two or three independent experiments. Co-IP was performed with mitotic MEFs that were immortalized by expression of SV40 large T antigen as previously described (*Baker et al., 2013*). Primary antibodies used were mouse anti-BubR1 (BD Transduction; as above), rabbit anti-mouse BubR1 (aa382-420) ([*Baker et al., 2004*]; as above), rabbit anti-Cdc20 (Santa Cruz; as above), mouse anti-Mad2 (BD Transduction, 610679, 1:1000), rabbit anti-Mad2 ([*Ricke et al., 2011*]; 1:1000). All antibodies were detected with secondary HRP-conjugated goat anti-mouse or anti-rabbit antibodies (Jackson Immunoresearch; as above) except when Cdc20 immunoblot was performed from CDC20 IP, in which Rabbit TrueBlot Anti-Rabbit IgG HRP (Rockland, Limerick, PA; 18-8816-33 1:1000) was used.

### Karyotype analyses

MEF karyotype analyses were performed as previously described on at least *n* = 3 individual MEF lines per genotype (*Babu et al., 2003*). Interphase FISH analysis on single cells isolated from various fresh tissues from 3-mo-old mice and Kras$^{LA1}$ hyperplastic lungs was performed as described previously (*Baker et al., 2008*), and were analyzed in the Mayo Clinic Cytogenetics Core Facility. At least 100 cells were analyzed per sample. At least *n* = 3 individual mice per genotype per tissue were used.

### Live-cell imaging experiments

Chromosome segregation analysis was performed on MEFs stably expressing H2B-RFP, as previously described (*Malureanu et al., 2009*). In mitotic timing experiments, the time interval between nuclear envelope breakdown (NEBD) and anaphase onset was measured in H2B-mRFP positive cells by monitoring unchallenged mitoses. Briefly, cells undergoing NEBD were marked and monitored at two minute intervals until anaphase onset. For SAC sensitivity experiments, cells were treated with

nocodazole (Sigma-Aldrich) at a final concentration of either 20 or 10 ng/ml and then monitored from NEBD to anaphase onset. Nocodazole challenge experiments were performed as previously described (*Malureanu et al., 2009*). Briefly, nocodazole was added to a final concentration of 100 ng/ml. Cells undergoing NEBD were marked and monitored at 10 min intervals to determine when they decondensed their chromosomes. The duration of arrest in mitosis, which is defined as the interval between NEBD (onset of mitosis) and chromatin decondensation (exit from mitosis without cytokinesis), was then calculated and plotted. For checkpoint silencing and sensitivity experiments, 500 nM or 2 μM AZ3146 (Sigma-Aldrich) was added either concurrently or in sequence with nocodazole. All experiments were performed on at least three independently generated MEF lines unless stated.

## Monastrol washout

Monastrol washout was performed as previously described (*Ricke et al., 2012*). Briefly 100 μM monastrol (Enzo Life Sciences, Famingdale, NY) was added to cells for 60 min, after which, 10 μM MG132 (Sigma-Aldrich) was added for 60 min. Cells were then released for 90 min into 10 μM MG132 alone before fixation (4% PFA for 10 min) and staining with Hoechst. Cells treated with Aurora B inhibitor were cultured in medium with 10 nM or 50 nM AZD1152-HQPA (ChemieTek, Indianapolis, IN), as specified for each experiment. Cells in which one or more chromosome was misaligned were considered misaligned. All experiments were performed on at least three independently generated MEF lines unless otherwise stated.

## Immunofluorescence

Immunofluorescence was performed and quantified as previously described (*Kasper et al., 1999*). In all cases, DNA was visualized with Hoechst and centromeres were visualized with human anti-centromeric antibody (Antibodies, Inc, Davis, CA; 15-234-001, 1:100). Primary antibodies used were mouse anti-BubR1 (BD Transduction; 612503, 1:250), rabbit anti-Flag (Cell Signaling; 2368S, 1:100), rabbit anti-Mad2 ([*Ricke et al., 2011*]; 1:500), and mouse-anti-PP2A-B56α (BD Transduction; 610615, 1:200). A laser-scanning microscope (LSM 880; Carl Zeiss) with an inverted microscope (Axiovert 100 M; Carl Zeiss) was used to capture images. For quantification, we used ImageJ software (National Institutes of Health, Bethesda, MD) as previously described (*Ricke et al., 2012*). All confocal microscopic images are representative of at least three independent experiments. All experiments were performed on at least three independently generated MEF lines.

## Statistical analyses

Prism software (GraphPad Software) was used for all statistical analyses. A two-tailed Mann-Whitney test was used for pairwise significance analysis in *Figure 2C*; *Figure 9A*; *Figure 9—figure supplement 1C*; *Figure 10B*. A log-rank Mantel-Cox test was used for significance analysis in *Figure 5B*; *Figure 9B*; *Figure 10C*. A two-tailed unpaired *t*-test was used for comparisons in the following figures: *Figure 2D*; *Figure 3B and D*; *Figure 4B–D*; *Figure 5A*; *Figure 5—figure supplement 1B*; *Figure 7B*; *Tables 1–3*. A two-tailed paired *t*-test was used for significance analysis in *Figure 9C*. A Fischer's exact two-tailed test was used for significance analysis in *Figure 10D and E*. For consistency in these analyses, significance is indicated as follows: *$p < 0.05$; **$p < 0.01$; and ***$p < 0.001$. Sample sizes were chosen based on previously published studies where differences were observed. No samples were excluded.

## Acknowledgements

We thank Liviu Malureanu, for initiating the project and the members of the van Deursen lab for helpful discussions and feedback. We thank Qianqian Guo for animal breeding and genotyping. This work was supported by grants from the National Institutes of Health: RMN (F30 CA189339) and JMvD (R01 CA096985 and R01 CA126828).

## Additional information

### Funding

| Funder | Grant reference number | Author |
|---|---|---|
| National Institutes of Health | CA096985 | Jan M van Deursen |
| National Institutes of Health | CA189339 | Ryan M Naylor |
| National Institutes of Health | CA126828 | Jan M van Deursen |

The funders had no role in study design, data collection and interpretation, or the decision to submit the work for publication.

### Author contributions

RLW, Conception and design, Acquisition of data, Analysis and interpretation of data, Drafting or revising the article, Contributed unpublished essential data or reagents; JFL, Acquisition of data, Analysis and interpretation of data, Drafting or revising the article; RMN, JMvD, Conception and design, Analysis and interpretation of data, Drafting or revising the article; KBJ, DJB, Acquisition of data, Drafting or revising the article

### Author ORCIDs

Jan M van Deursen, http://orcid.org/0000-0002-3042-5267

### Ethics

Animal experimentation: This study was performed in strict accordance with the recommendations in the Guide for the Care and Use of Laboratory Animals of the National Institutes of Health. All protocols were reviewed and approved by the Mayo Clinic Institutional Animal Care and Use Committee (protocol A48013).

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
