## [Decision Letter]

Thank you for submitting your article "*BubR1* alterations that reinforce mitotic surveillance act against aneuploidy and cancer" for consideration by *eLife*. Your article has been favorably evaluated by Sean Morrison as the Senior editor and three reviewers, one of whom, Yukiko M Yamashita, is a member of our Board of Reviewing Editors.

The reviewers have discussed the reviews with one another and the Reviewing Editor has drafted this decision to help you prepare a revised submission.

Summary:

In this study, van Deursen and colleagues show that the *BubR1* mutant lacking the middle region (ΔI) is fully functional in suppressing Kras-driven lung cancer formation and associated aneuploidy in mice, whereas either *BubR1*^ΔN^ or *BubR1*^N^ is not functional. Strikingly, *BubR1*^ΔI^ is more active that the wild type (WT) in establishing or supporting the checkpoint in cultured cells. These results definitively show that the middle region of BubR1 has a negative function in the spindle checkpoint, and are generally consistent with the findings of Diaz-Martinez et al. which showed that human BubR1 with the middle region deleted was more active than BubR1 WT in human cells.

The appeal of this paper is that the authors are conducting these experiments primarily in mice, compared to most of the previous work that was performed largely in tissue culture cells. Thus, the results presented in this manuscript were considered to be highly significant. However, at the same time, it was discussed that this work is a direct extension of their previous paper looking at a limited set of *BubR1* mutants for the phenotypes that they previous reported. In addition, this paper does not provide any molecular insights into the function of BubR1 in the spindle assembly checkpoint or chromosome segregation.

Taken together, the reviewers agreed that it will be important for the authors to provide at least some degree of mechanistic insights (described below). The reviewers have discussed a few options so as to avoid excessive revisions.

Essential revisions:

1) Results in Figure 6 actually suggest that there is more Mad2-Cdc20 binary complex in *BubR1*^ΔI^ cells, relative to intact MCC. It is unclear why this would lead to a hyperactive checkpoint, as Mad2-Cdc20 is a weaker APC/C inhibitor compared to MCC. One possibility is that *BubR1*^ΔI^ overexpression affects Cdc20 phosphorylation by Bub1 and Plk1, thereby inhibiting APC/C-Cdc20 in an MCC-independent manner. The authors should examine the phosphorylation status of Cdc20. Site-specific Cdc20 antibodies are available for human Cdc20. These might work for mouse Cdc20. If not, phos-tag gels might reveal gel mobility shift of phospho-Cdc20.

2) Additionally, the reviewers felt that it will be much more informative if the authors examine the phenotypes of *BubR1* mutants with Phe, D2, and KARD individually deleted, which will add molecular mechanism to this manuscript beyond what has been shown by their own earlier work. These experiments should be performed in culture in MEFs.

---

## [Author Response]

*Taken together, the reviewers agreed that it will be important for the authors to provide at least some degree of mechanistic insights (described below). The reviewers have discussed a few options so as to avoid excessive revisions.*

Essential revisions:

*1) Results in Figure 6 actually suggest that there is more Mad2-Cdc20 binary complex in BubR1^ΔI^ cells, relative to intact MCC. It is unclear why this would lead to a hyperactive checkpoint, as Mad2-Cdc20 is a weaker APC/C inhibitor compared to MCC. One possibility is that BubR1^ΔI^ overexpression affects Cdc20 phosphorylation by Bub1 and Plk1, thereby inhibiting APC/C-Cdc20 in an MCC-independent manner. The authors should examine the phosphorylation status of Cdc20. Site-specific Cdc20 antibodies are available for human Cdc20. These might work for mouse Cdc20. If not, phos-tag gels might reveal gel mobility shift of phospho-Cdc20.*

We agree with the reviewers’ observation that there are more Mad2-Cdc20 binary complexes in *BubR1*^ΔI^ MEFs and we were also intrigued by this result. We further validated this finding by performing an immunoprecipitation of Cdc20 and probing for Mad2, demonstrating that there is indeed increased binding between Cdc20 and Mad2 in *BubR1*^ΔI^ MEFs. This new result has been incorporated into revised Figure 6 of the manuscript and have been described in the first paragraph of the subsection “The mitotic checkpoint complex composition is unique in *BubR1*^ΔI^ MEFs”.

Additionally, we performed the proposed reviewer experiments of evaluating the phosphorylation status of two key residues of Cdc20 in FL-*BubR1* and *BubR1*^ΔI^ MEFs (S153 and S92 by Bub1 kinase and Plk1 kinase respectively) to determine if MCC-independent mechanism were contributing to heightened SAC activity. We found the indicated antibodies were indeed able to detect murine Cdc20. However, we did not find an increase in phosphorylation of these residues, suggesting this signaling pathway is not hyperactivated in our mutants. These data can be found in revised Figure 6 of the manuscript and have been incorporated in the text in the last paragraph of the subsection “The mitotic checkpoint complex composition is unique in *BubR1*^ΔI^ MEFs”.

We also analyzed the phosphorylation status of Cdc20 by Phos-tag western blotting, which confirmed that phosphorylation-mediated inhibition of Cdc20 is unchanged in *BubR1*^ΔI^ MEFs. Please see Figure 11.

Author response image 1.**DOI:**
http://dx.doi.org/10.7554/eLife.16620.034

While it is counterintuitive that MEFs overexpressing *BubR1*^ΔI^ show more robust SAC signaling than those overexpressing FL-*BubR1* given the substantial difference in MCC formation, *BubR1*^ΔI^ MEFs are unique in that they are sensitive to mild perturbations in microtubule-kinetochore attachment caused by low concentrations of nocodazole. Therefore, one potential explanation is that a decreased threshold for activation or sustainability of SAC signaling (or both) affords *BubR1*^ΔI^ MEFs a more robust SAC. We have now more clearly discussed this potential explanation in the revised manuscript. Please see the first paragraph of the subsection “The mitotic checkpoint complex composition is unique in *BubR1*^ΔI^ MEFs”.

*2) Additionally, the reviewers felt that it will be much more informative if the authors examine the phenotypes of BubR1 mutants with Phe, D2, and KARD individually deleted, which will add molecular mechanism to this manuscript beyond what has been shown by their own earlier work. These experiments should be performed in culture in MEFs.*

We agree with the reviewers that a fundamental limitation of our manuscript was a lack of refinement of the selected domains of BubR1 for deletion, and we have performed the experiments requested of the reviewers to gain additional mechanistic insight. To this end, we generated the following doxycycline-inducible lentiviral *BubR1* deletion constructs: FL-*BubR1* and *BubR1*^ΔI^ to serve as controls for the original transgenic MEFs; *BubR1*^ΔPhe^ (lacking the Phe box); *BubR1*^ΔD^ (lacking D-box2); *BubR1*^ΔPheD^ (lacking both the Phe box and D-box2); and *BubR1*^ΔKARD^ (lacking the KARD). We expressed these constructs in wild-type MEFs and upon induction with doxycycline, found them to be highly overexpressed relative to endogenous *BubR1*. With these MEFs, we performed experiments focused on the domains’ contribution to error correction, mitotic timing, and spindle assembly checkpoint sustainability.

We found that all of our mutant constructs were able to prevent misalignments resulting from Aurora B deficiency, suggesting that all regions within *BubR1*^ΔI^ are dispensable for error correction. On the other hand, the gain of function of a robust spindle assembly arrest caused by overexpression of *BubR1*^ΔI^ seemed dependent on the loss of both the Phe box and D-box2, as only *BubR1*^ΔPheD^ recapitulated this phenotype. This aligns with previous work from the laboratory of Dr. Hongtao Yu suggesting these regions serve to normally interfere with checkpoint signaling, especially in combination. We have incorporated this information into revised Figure 8, revised Figure 8—figure supplement 1,revised Figure 9, and at various places in the text. Please see the last paragraph of the Introduction and the subsection “Refined *BubR1*^ΔI^ mutants are capable of reinforcing error correction and SAC signaling”.

Finally, we found that the increase in mitotic timing was exclusive to our *BubR1*^ΔI^ mutant, as no other mutants had this impact. We generated a compound mutant (*BubR1*^ΔPheDΔKARD^) lacking the Phe box, D-box2 and KARD that also had no impact on timing when overexpressed (revised Figure 9—figure supplement 1). This offers the intriguing possibility that there may be a region within *BubR1* between the Phe/D-box2 and KARD contributing to timing that has yet to be mapped.

We have incorporated all these data in two new figures, revised Figure 8 and Figure 9, and two supplementary figures, revised Figure 8—figure supplement 1 and revised Figure 9—figure supplement 1. For description and interpretation and discussion of the data, please see the Abstract,, the last paragraph of the Introduction, the first two paragraphs of the Discussion, and the fifth and sixth paragraphs of the Discussion).